# Circadian RNA expression elicited by 3′-UTR IRAlu-paraspeckle associated elements

**Manon Torres, Denis Becquet, Marie-Pierre Blanchard, Séverine Guillen, Bénédicte Boyer, Mathias Moreno, Jean-Louis Franc, Anne-Marie François-Bellan\***

Faculté de Médecine Nord, Aix Marseille Université, CNRS, CRN2M-UMR7286, Marseille, France

**Abstract** Paraspeckles are nuclear bodies form around the long non-coding RNA, Neat1, and RNA-binding proteins. While their role is not fully understood, they are believed to control gene expression at a post-transcriptional level by means of the nuclear retention of mRNA containing in their 3′-UTR inverted repeats of Alu sequences (IRAlu). In this study, we found that, in pituitary cells, all components of paraspeckles including four major proteins and Neat1 displayed a circadian expression pattern. Furthermore the insertion of IRAlu at the 3′-UTR of the EGFP cDNA led to a rhythmic circadian nuclear retention of the egfp mRNA that was lost when paraspeckles were disrupted whereas insertion of a single antisense Alu had only a weak effect. Using real-time video-microscopy, these IRAlu were further shown to drive a circadian expression of EGFP protein. This study shows that paraspeckles, thanks to their circadian expression, control circadian gene expression at a post-transcriptional level.

*For correspondence: anne-marie.francois@univ-amu.fr

**Competing interests:** The authors declare that no competing interests exist.

## Introduction

The circadian clock orchestrates daily rhythms in metabolism, physiology and behavior that allow organisms to anticipate regular changes in their environment, increasing their adaptation (*Asher and Schibler, 2011*). Circadian rhythms are underpinned by daily rhythms of gene expression. The transcriptional component of these rhythms is well understood (*Zhang and Kay, 2010*). The circadian variation in abundance of the positive (Clock and Bmal1) and the negative (Per1, Per2 and Cry1, Cry2) components of these loops drive the circadian transcription of both direct targets genome-wide and a cascade of circadian output transcription factors, which together mediate the circadian transcriptional profile of a cell type or tissue (*Asher and Schibler, 2011*; *Rey et al., 2011*). Though the core circadian system has concentrated on transcriptional control, it has been apparent that substantial regulation is achieved after transcription so that post-transcriptional controls are emerging as crucial modulators of circadian clocks (*Lim and Allada, 2013*; *Wang et al., 2013*; *Menet et al., 2012*; *Koike et al., 2012*; *Hurley et al., 2014*). While in eukaryotes, approximately 1%-10% genes are subjected to circadian control directly or indirectly only ~1/5 of the mRNAs that display rhythmic expression are directly driven by transcription, which suggests that post-transcriptional mechanisms including RNA splicing, polyadenylation, mRNA stability, mRNA cytoplasmic export and RNAs nuclear retention are essential layers for generation of gene expression rhythmicity (*Partch et al., 2014*; *Menet et al., 2012*; *Koike et al., 2012*; *Hurley et al., 2014*).

Paraspeckles are recently identified nuclear bodies that have been shown to retain RNAs in the nucleus. Paraspeckles contain proteins PSPC1, RBM14, NONO, and SFPQ (*Prasanth et al., 2005*) and are usually detected as a variable number of discrete dots found in close proximity to nuclear speckles (*Bond and Fox, 2009*). One long noncoding RNA, nuclear-enriched abundant transcript

**eLife digest** Many biological features of animals, including body temperature and hormone levels, follow daily rhythms that repeat every 24 hours. These so-called circadian rhythms are driven by an internal body clock and are essential for the organism to adapt to the daily cycle of light and dark. Circadian rhythms also take place inside individual cells – for example, the amount of a given protein in a cell often rises and falls over each 24-hour period.

To generate these daily fluctuations, the processes used to make proteins based on the instructions encoded within a gene must be carefully controlled. Genes are first copied or 'transcribed' into intermediate molecules called messenger RNAs (mRNAs). These mRNA molecules must then travel out of the cell's nucleus before they can be de-coded to produce proteins. This means that daily fluctuations in mRNA and protein levels could occur because the rate at which the DNA is transcribed fluctuates or because controlling the steps that occur after transcription. However it is not clear how much these post-transcriptional steps contribute to circadian rhythms inside cells.

Recently, structures called paraspeckles were seen inside the nucleus. These structures are made from a long RNA molecule that does not code for a protein, and a number of proteins that can bind mRNA molecules. Paraspeckles are thought to prevent certain mRNAs from leaving the nucleus and therefore stop them from being decoded to make proteins. Torres et al. have now investigated whether paraspeckles may play a role in circadian rhythms.

Torres et al. looked at the long non-coding RNA and several proteins that are known to be components of paraspeckles in cells taken from the pituitary glands of rats using a variety of techniques. These cells were chosen because they were known to have a working circadian clock. The analysis showed that the levels of these components, as well as the number of paraspeckles within the nucleus, changed over the course of a daily cycle.

Torres et al. then confirmed that mRNAs containing a sequence that is known to recruit mRNAs to paraspeckes (the IRAlu sequence) could be also retained in the nucleus or released with a circadian rhythm. This pattern was lost when the paraspeckles were disrupted.

These findings suggest that daily fluctuations in protein levels can be post-transcriptionally controlled by paraspeckles rhythmically retaining mRNAs in the nucleus. Future studies could explore whether it may be possible to control circadian rhythms by targeting the paraspeckles, which could help to improve conditions where the internal body clock goes wrong.

one (Neat1), exclusively localized to paraspeckles serves as a structural component (*Hutchinson et al., 2007*; *Chen and Carmichael, 2009*; *Clemson et al., 2009*; *Sasaki and Hirose, 2009*; *Sunwoo et al., 2009*). The locus generates short and long transcripts from the same pro-moter, which have previously been identified as MENε (Neat1-1) and MENβ (Neat1-2), respectively (*Guru et al., 1997*; *Sasaki and Hirose, 2009*). Because specific depletion of Neat1-2 leads to disruption of paraspeckles (*Sasaki and Hirose, 2009*) Neat1-1 alone cannot induce paraspeckle formation.

Paraspeckles have been shown to retain in the nucleus RNAs containing duplex structures (*Chen and Carmichael, 2008*). This is the case for the mouse cationic amino acid transporter 2 (Cat2) transcribed nuclear RNA, Ctn-RNA, an alternatively spliced form of the Cat2 mRNA, which contains a dsRNA structure resulting from inverted short inter-spersed nuclear elements (SINEs) in its 3'-UTR (*Prasanth et al., 2005*). In human cells, hundreds of genes contain inverted repeated SINEs (mainly Alu elements) in their 3'-UTRs. Alu elements are unique to primates and account for almost all of the human SINEs and >10% of the genome. Their abundance leads to the frequent occurrence of inverted repeat structures (inverted repeated Alu elements [IRAlus]) in gene regions (*Chen et al., 2008*). It has been reported previously that mRNAs containing IRAlus in their 3'-UTRs like Nicolin 1 (NICN1) or Lin 28 are retained in the nucleus in paraspeckles (*Chen et al., 2008*; *Chen and Carmichael, 2008*). Therefore, this nuclear retention pathway of IRAlus in 3'-UTRs of genes provides an additional layer of gene regulation by sequestering mature mRNAs within the nucleus.

We recently reported that two protein components of paraspeckles, namely NONO and SFPQ, display a circadian expression pattern in primary cultures of pituitary cells as well as in a rat pituitary cell line, the GH4C1 cells (*Becquet et al., 2014*; *Guillaumond et al., 2011*). We used this cell line to determine whether one of the posttranscriptional mechanisms allowing circadian gene expression in pituitary cells could involve the circadian nuclear mRNA retention by paraspeckle bodies. To this end, we first characterized the presence of paraspeckles and we showed that these nuclear bodies were rhythmically expressed in the rat GH4C1 pituitary cell line. We then made a series of EGFP-fused IRAlu or Alu constructs and transfected them into GH4C1 cells to investigate the EGFP expression and the fates of the egfp-IRAlu or egfp-Alu RNA. We showed that an IRAlu element in the 3'-UTR of the egfp mRNA strongly repressed EGFP expression. Further, this reduction was accompanied by significant nuclear retention of the mRNAs, likely by paraspeckle bodies. We showed also that insertion of IRAlus in the 3'-UTR of EGFP reporter gene allowed rhythmic nuclear egfp-IRAlu RNA retention and rhythmic EGFP protein expression. Finally, this rhythmic nuclear egfp-IRAlu RNA retention as well as the rhythmic nuclear retention of some known cycling transcripts that were shown here associated with paraspeckles was lost when paraspeckles were disrupted.

## Results

### Characterization of paraspeckle nuclear bodies in pituitary GH4C1 cells

#### Visualization of paraspeckle protein components by immunofluorescence and biochemical evidence for their association with the long noncoding RNA Neat1

The presence of paraspeckles in GH4C1 cells was anatomically evidenced by confocal microscopy on the basis of the overlap of their protein components using antibodies directed against SFPQ, NONO, PSPC1 and RBM14. SFPQ (green *Figure 1A,C*), NONO (green *Figure 1B,D*), PSPC1 (red *Figure 1A,B*) and RBM14 (red *Figure 1C,D*) staining appeared as punctate clusters located exclusively within the boundaries of the nucleus as delimited by Hoechst labeling (*Figure 1A,B,C,D*, 4th column). When SFPQ or NONO staining was merged with PSPC1 (*Figure 1A,B*, 3rd and 4th columns) on the one hand or RBM14 (*Figure 1C,D*, 3rd and 4th columns) on the other hand, some punctate clusters were found to overlap with each other in subnuclear structures reminiscent of paraspeckles. The protein components of paraspeckles were further shown to be associated with the long noncoding RNA Neat1lncRNA Neat1, using RNA-immunoprecipitation (RIP) experiments with antibodies directed against NONO, SFPQ, RBM14 or PSPC1. Primers used in RT-qPCR detected both long (Neat1-2) and short (Neat1-1) isoforms of Neat1 RNA (*Figure 2—source data 1*). A 20- to 80-fold enrichment in Neat1 RNA was obtained with these antibodies as compared to an irrelevant non-specific antibody (*Figure 2A*). These results showed that the four major protein components of paraspeckles bound the structural lncRNA Neat1, confirming the presence of paraspeckle nuclear bodies in GH4C1 cells.

#### Visualization of paraspeckle nuclear bodies by fluorescence in situ hybridization (FISH) of long noncoding RNA Neat1

Paraspeckle nuclear bodies were then visualized using Neat1 RNA staining by FISH. Under confocal laser scanning microscope, Neat1 RNA FISH staining appeared as regular punctates within the boundaries of the nucleus (*Figure 2B*). Number of foci was low and variable from one cell to the other. To gain resolution in the intranuclear spatial arrangement of Neat1 RNA, we combined RNA FISH with Stochastic Optical Reconstruction Microscopy (STORM). Under conventional fluorescence microscopy, resolution of paraspeckle bodies was very low as can be seen in the left part and in upper panel of the right part (red staining) in *Figure 2C*. The increase in resolution from conventional to super-resolution microscopy was apparent in the upper panel of the right part of *Figure 2C* in which the high-resolution image obtained after STORM analysis (white staining) was superimposed to the conventional fluorescence image (red staining). Due to the poor resolution, the size of the paraspeckles was imprecise and could not be measured under conventional fluorescence microscope (red staining). By contrast, dimensions could be precisely measured under STORM (bottom panel of the right part in *Figure 2C*). In addition to the precision in the real size of paraspeckle bodies,

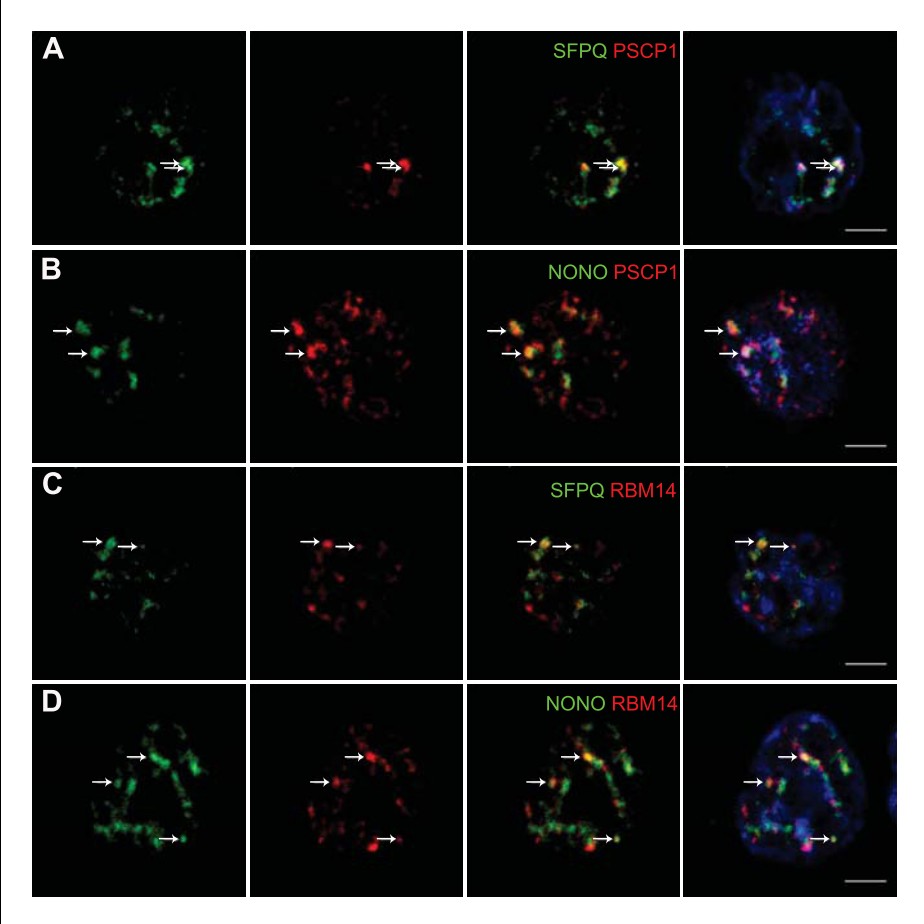

**Figure 1.** Localization by confocal microscopy of paraspeckle proteins in GH4C1 pituitary cells. Cells are grown on coverslips and labeled for immunofluorescence with antibody to SFPQ (green **A**, **C**), NONO (green **B**, **D**) PSPC1 (red **A**, **B**) or RBM14 (red **C**, **D**). SFPQ or NONO staining is merged with PSPC1 (**A**, **B** 3th column) or RBM14 (**C**, **D** 3th column). Nuclear staining by Hoechst for the same samples is added in 4th column. Arrows indicate punctate clusters in which two paraspeckle proteins overlap. Scale bars: 5 µm.

super-resolution analysis allowed also to show that Neat1 labeling of paraspeckle bodies was not round in shape, being merely elliptical. Indeed using an imaging software (NIS-Elements, Nikon France S.A, Champigny sur Marne, France) we showed that the averaged ratio of Height to Width (n=10) was >1 (1.53 ± 0.1) and the mean surface area was 24,960 ± 4831 nm$^2$ (*Figure 2C*).

## Rhythmic expression of paraspeckle components in GH4C1 cells

### Rhythmic expression of protein components

To determine whether paraspeckle nuclear bodies displayed a circadian expression pattern we looked for rhythmic expression of their protein components as determined by Western blot analysis in nuclear protein extracts. We previously reported that the two proteins NONO and SFPQ followed a rhythmic expression pattern in GH4C1 cells (*Guillaumond et al., 2011*). It holds also true for two other paraspeckle-associated proteins, RBM14 and PSPC1, as reported in *Figure 3A*. Indeed, RBM14 and PSPC1 proteins displayed a rhythmic pattern in GH4C1 cells over the T2-T30 time period that could be fitted with a non-linear cosinor fit in which the period value (2pi/Frequency) was constrained to the circadian period value 24 hr (equation values given in *Figure 3—source data 1*). It may be noticed that the rhythmic expression pattern of RBM14 and PSPC1 proteins (*Figure 3A*) matched quite correctly with that of NONO and SFPQ proteins we previously reported (see Figure 6A in *Guillaumond et al., 2011*).

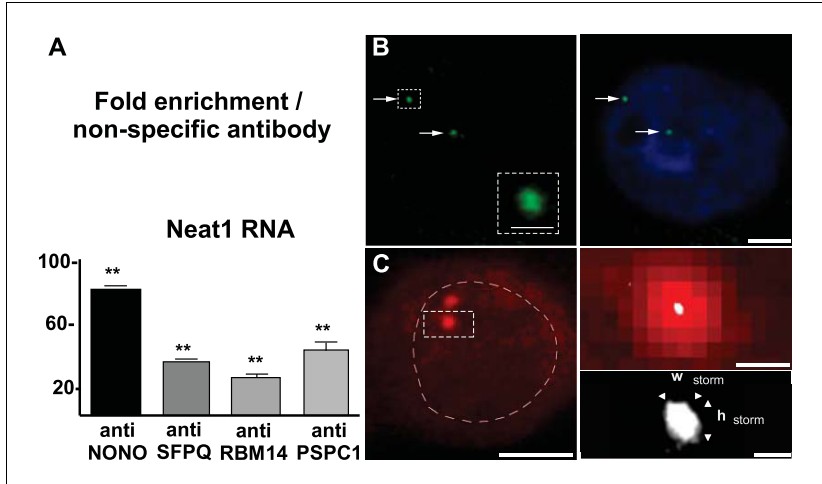

**Figure 2.** Neat1 RNA in paraspeckle nuclear bodies: association with paraspeckle proteins and visualization by FISH. (**A**) Association of paraspeckle proteins with Neat1 RNA RNA Immuno-Precipitation (RIP) experiments (n=4 for each antibody) using antibodies directed against NONO, SFPQ, RBM14 or PSPC1 show the enrichment in Neat1 RNA obtained as compared to an irrelevant non-specific antibody (see *Figure 2—source data 1* for primer sequences). **p<0.01 *vs* non-specific antibody. B. Visualization of Neat1 RNA by Fluorescence in situ Hybridization and confocal laser scanning microscope **Left Panel:** RNA-FISH shows the distribution of Neat1 RNA in a few distinct foci (arrows). The round aspect of the foci under confocal laser scanning microscope is shown in the insert in which assigned foci is enlarged. Scale bars: 1 μm **Right Panel:** The extent of the nucleus is shown with Hoechst staining. Foci containing Neat1 RNA localize within the nucleus sometimes in the close vicinity of nucleus boundaries. Scale bars: 5 μm. (**C**) Visualization of Neat1 RNA by Fluorescence in situ Hybridization and super resolution **Left Panel:** Conventional fluorescence microscopy of Neat1 RNA-FISH. The nucleus is outlined with hand-drawn dashed lines to indicate the nuclear periphery. Scale bars: 5 μm. **Right Upper Panel:** Enlargement of the foci assigned in left panel allows to show the poor resolution of paraspeckle under conventional fluorescence microscopy (in red). In white is the superimposed high-resolution image obtained after STORM analysis. Note that the size of paraspeckle after STORM analysis is strongly reduced. Scale bars: 0.5 μm. **Right Bottom Panel:** Enlargement of the STORM analysis shown in the upper panel with measurements of width (Wstorm=0.14 μm) and height (hstorm=0.17 μm). Note the elliptical shape of the foci analyzed. Scale bars: 0.1 μm.

The following source data is available for figure 2:

**Source data 1.** Sequences of qPCR primers and oligonucleotides.

## Rhythmic expression of Neat1 RNA

We further looked for a circadian expression pattern of the structural element of paraspeckles, namely the lncRNA Neat1. As shown in *Figure 3B*, Neat1 RNA levels (Neat1-1 + Neat1-2) displayed a rhythmic pattern in GH4C1 cells over the T2-T38 time period (cosinor fit values given in *Figure 3—source data 1*). Neat1 RNA levels (Neat1-1 + Neat1-2) also displayed a circadian expression pattern in several mouse tissues including the pituitary gland but also other peripheral oscillators such as the spleen or the adrenal gland (*Figure 3—figure supplement 1*). This was also the case in the central clock, namely the suprachiasmatic nuclei (*Figure 3—figure supplement 1*).

## Rhythmic binding of paraspeckle proteins on Neat1 RNA

We used RIP experiments to examine whether binding of NONO, SFPQ, RBM14 and PSPC1 on Neat1 RNA in GH4C1 cells was rhythmic. As shown in *Figure 3C*, binding of the four paraspeckle-associated proteins on Neat1 RNA displayed a rhythmic pattern (cosinor fit values given in *Figure 3—source data 2*). Maximum binding on Neat1 RNA was reached between 6 and 10 hr after the medium change for the four proteins (*Figure 3C*).

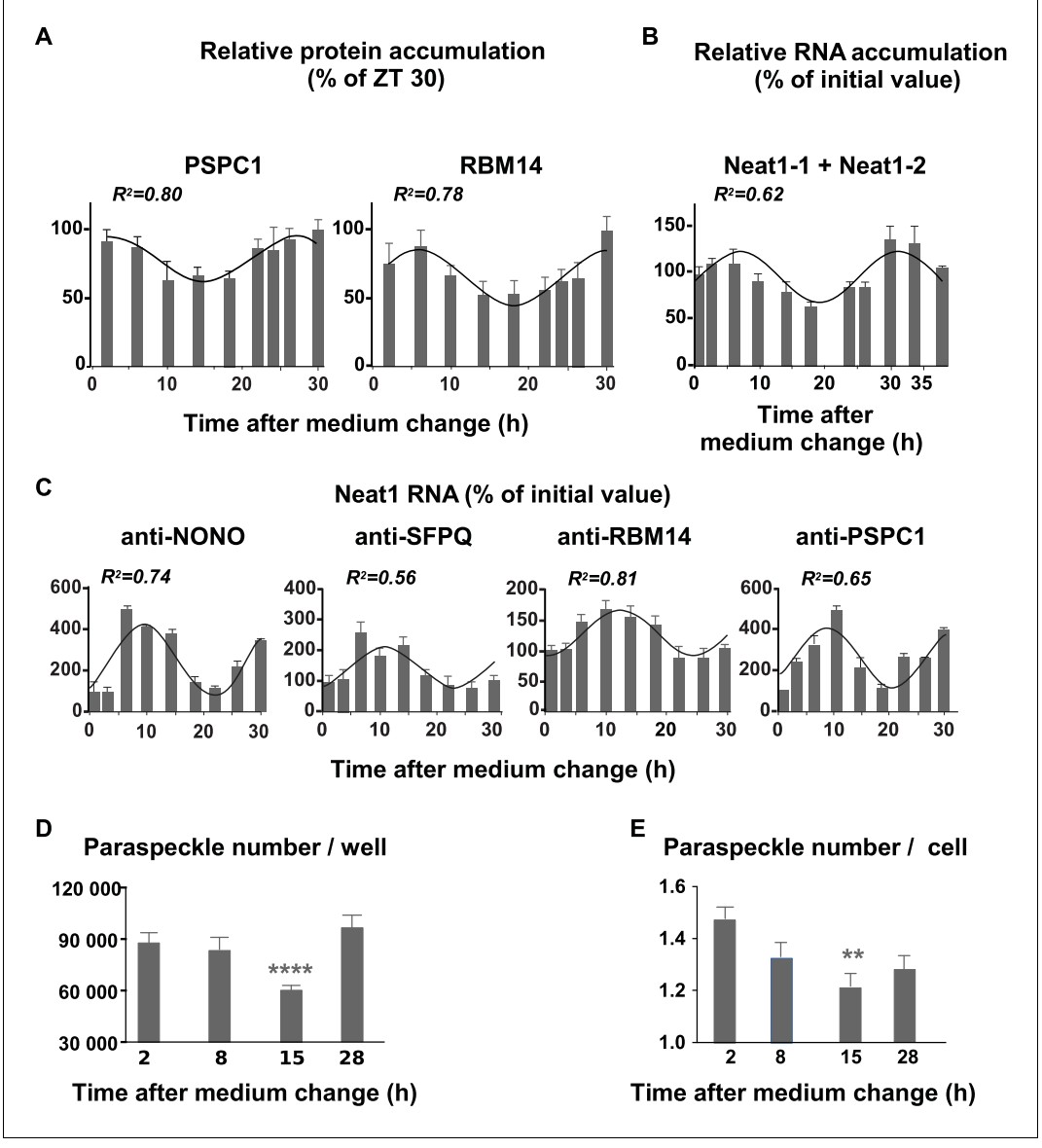

**Figure 3.** Rhythmic expression and association of paraspeckle components. (**A**) Rhythmic expression of two paraspeckle proteins in pituitary GH4C1 cells The expression of PSPC1 and RBM14 is determined by Western Blot analysis over a 30 hr time period. Each data point (mean ± SD of three independent samples) represents the ratio of the depicted proteins to ATF2 and is expressed relative to the value obtained at ZT 30. Experimental values can be adequately fitted ($R^2 > 0.55$) with a non-linear cosinor equation in which the period value is set to 24 hr (see also *Figure 3—source data 1*). (**B**) Rhythmic expression of the long noncoding Neat1 RNA. The expression of the lncRNA Neat1 is determined by RT-qPCR over a 40 hr time period. Primers used to allow the detection of both Neat1-1 and Neat1-2. Experimental values (n=4) expressed as a percent of the initial value obtained at ZT 0 can be adequately fitted ($R^2 > 0.55$) with a non-linear cosinor equation in which the period value is set to 24 hr (see also *Figure 3—source data 1*). (**C**) Rhythmic association of paraspeckle proteins with Neat1 RNA RNA Immuno-Precipitation (RIP) experiments (n=4 for each antibody) are performed over a 30h time period. At each time point, the levels of Neat1 RNA determined after immuno-precipitation by the antibodies directed against NONO, SFPQ, RBM14 and PSPC1 were normalized relative to Neat1 RNA input levels and expressed as a percent of the value obtained at T0. Experimental values can be adequately fitted ($R^2 > 0.55$) with a non-linear cosinor equation in which the period value is set to 24 hr (see also *Figure 3—source data 2*). (**D–E**) Rhythmic fluctuations of paraspeckle number Cells were arrested at four different times after the medium change and processed for FISH of Neat1 RNA. At each time point, 20 to 35 images from four wells of 100 000 cells obtained in two different experiments were acquired under a confocal microscope with a 40X objective. At each time point, the total number of paraspeckles per well and the mean number of paraspeckles per cell were calculated. **p<0.001 ****p<0.0001.
*Figure 3 continued on next page*

*Figure 3 continued*

The following source data and figure supplement are available for figure 3:

**Source data 1.** Cosinor analysis of the rhythmic expression pattern of paraspeckle components in GH4C1 cells.
**Source data 2.** Cosinor analysis of the rhythmic binding of the four paraspeckle-associated proteins on Neat1 RNA in GH4C1 cells.
**Figure supplement 1.** Rhythmic expression of the long noncoding Neat1 RNA in diverse circadian oscillators.

## Rhythmic number of paraspeckles

In cells arrested at four different times after the medium change, the total number of paraspeckles per well was shown to fluctuate, being significantly lower 15 hr after the medium change i.e. at a time when Neat1 RNA levels were around the lowest levels (*Figure 3D*, $F_{3,103}=9.531$ p<0.0001). Furthermore at this time point, the mean number of paraspeckles per cell reached a minimum value (*Figure 3D*, $F_{3447}= 5.456$, p<0.001). Taken together, these results showed that the rhythm of Neat1 and its associated proteins reported above translate into a rhythm in the number of paraspeckles inside the cells.

## Influence of IRAlu elements inserted in 3'-UTR EGFP mRNA

### IRAlu elements reduced EGFP protein expression

Paraspeckles have been shown to retain in the nucleus RNAs containing duplex structures from inverted repeats of the conserved Alu sequences (IRAlu elements) within their 3'-UTR (*Chen and Carmichael, 2008*). This has been shown to be the case for Nicolin 1 (NICN1) gene (*Chen and Carmichael, 2008*). We utilized EGFP expression reporter system to investigate the effects of IRAlu from the 3'-UTR of NICN1 gene. The single antisense Alu, or the IRAlu elements cloned from the 3'-UTR of NICN1 were inserted each between the EGFP cDNA 3'-UTR region and the SV40 polyadenylation signal of the expression vector pEGFP-C1 to generate constructs that were then stably transfected into GH4C1 cells. In agreement with previous results by Chen et al (*Chen and Carmichael, 2008*), the IRAlu elements derived from NICN1 significantly reduced EGFP expression when compared with the Alu element (*Figure 4A–C*). This is here evidenced both by the significant reduction in the number of fluorescent cells (*Figure 4B*) and by the significant decrease in relative EGFP levels measured by western blotting in IRAlu-egfp cell line compared to both Alu–egfp cell line (*Figure 4C*). By contrast, we have not observed any difference in gene expression between the parent plasmid pEGFP and derivatives that contained a single Alu element (data not shown).

### IRAlu element induced egfp mRNA nuclear retention

The reduced EGFP protein expression in IRAlu-egfp cell line suggested that IRAlu can induce a stronger nuclear retention of egfp mRNA than Alu did, as previously shown by Chen et al (*Chen and Carmichael, 2008*). Given that egfp mRNA cytoplasmic localization correlates strongly with EGFP expression, we confirmed in our cell lines that after fractionating cytoplasmic and nuclear RNAs, IRAlu-containing egfp mRNA appeared to be preferentially retained in the nucleus in comparison with Alu-containing egfp mRNA. As shown in *Figure 4D*, the IRAlu from Nicn1 caused a more than two-fold greater nuclear retention of the egfp mRNA when compared with the corresponding Alu element. On the other hand, there is no significant difference in nuclear/cytoplasmic distribution of egfp-Alu mRNA compared with control (data not shown), consistent with our finding that a single Alu element in the 3'-UTR does not affect EGFP gene expression. It then appears that nuclear retention of IRAlu-containing egfp mRNA correlates with silencing of EGFP protein expression (*Figure 4A–C*).

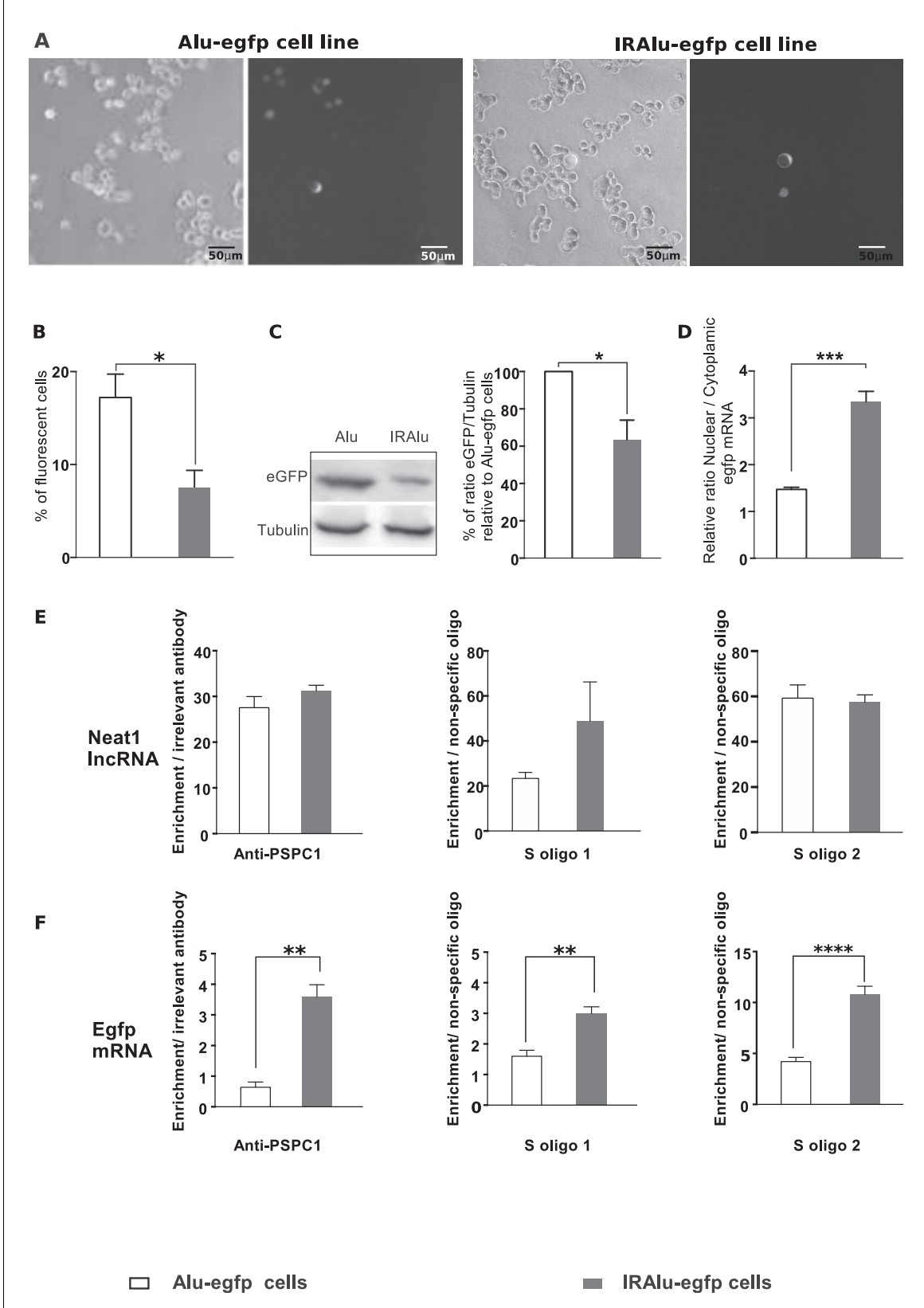

**Figure 4.** Influence of IRAlu elements inserted in 3'-UTR egfp mRNA. (**A–C**) Decrease in EGFP expression by insertion of IRAlus in the 3' -UTR of egfp mRNA. IRAlus and Alu were PCR-amplified from the 3'-UTR of Nicn1 and then inserted separately into the 3'-UTR of egfp mRNA. GH4C1 cell lines, Alu-

*Figure 4 continued on next page*

*Figure 4 continued*

egfp and IRAlu-egfp cell lines were established by transfection of the indicated plasmids. (A) Representative example of fluorescence and corresponding bright field pictures taken 48 hr after platting of each cell line. Scale bars equal 50 μm. (B) Quantitative analysis of the percent of fluorescent cells in each cell line. Data are means ± SEM of 18 measures performed in 3 independent experiments. (C) Quantification of relative levels of eGFP investigated by western blotting with anti-GFP antibody in total protein extracts from the two cell lines. Tubulin was used as the loading control. Data are mean ± SEM of values obtained from three experiments in IRAlu-egfp cell line and are expressed as a percent of the corresponding value obtained in Alu-egfp cell line. (D) Nuclear and cytoplasmic egfp mRNA were quantified by qPCR in each cell line and normalized to the relative amount of gapdh mRNA (n=8 for each cell line). Ratio of nuclear versus cytoplasmic egfp mRNA levels are compared between IRAlu-egfp and Alu-egfp cell lines. (E) Enrichment in lncRNA Neat1 after RNA Immuno-Precipitation (RIP) with an antibody directed against PSPC1 relative to an irrelevant antibody (left panel) or after RNA pull-down with two different specific biotinylated oligonucleotides (S oligo 1 and S oligo 2) relative to a non-specific oligonucleotide (two right panels). The relative enrichment in lncRNA Neat1 obtained after either RIP (n=3 for each cell line) or RNA pull-down (n=6 for each cell line) is not statistically different in Alu-egfp versus IRAlu-egfp cell lines F. Enrichment in egfp mRNA after RNA Immuno-Precipitation (RIP) with an antibody directed against PSPC1 relative to an irrelevant antibody (left panel) or after RNA pull-down with two different specific biotinylated oligonucleotides (S oligo 1 and S oligo 2) relative to a non-specific oligonucleotide (two right panels). The relative enrichment in egfp mRNA obtained after either RIP (n=3 for each cell line) or RNA pull-down (n=6 for each cell line) is statistically higher in IRAlu-egfp versus Alu-egfp cell lines. *p<0.05 **p<0.01***p<0.001****p<0.0001.

The following figure supplements are available for figure 4:

**Figure supplement 1.** Secondary structure of the first 2500 nucleotides of the Neat1 RNA as predicted by MaxExpect software.

**Figure supplement 2.** Enrichment of Neat1 RNA relative to input after RNA pull-down.

## IRAlu element associated with paraspeckle components

### IRAlu element associated with PSPC1 protein

We also asked whether IRAlu-containing egfp mRNA is associated with paraspeckle protein PSPC1. To answer this question, we performed RIP experiments on Alu-egfp and IRAlu-egfp cell lines using PSPC1 antibody compared to irrelevant antibody. While the enrichment in lncRNA Neat1 obtained after use of PSPC1 antibody was comparable in the two cell lines (*Figure 4E*), enrichment in egfp mRNA was significant only in IRAlu-egfp cell line (*Figure 4F*), attesting that PSPC1 protein was associated to IRAlu-egfp and not to Alu-egfp mRNA.

### IRAlu element associated with lncRNA Neat1

We next asked whether IRAlu-containing egfp mRNA is associated with endogenous lncRNA Neat1. To answer this question, we adapted a pull-down Neat1 technology from two published approaches (RIA (RNA-interactome analysis) and CHART (Capture Hybridization Analysis of RNA Targets) (*Kretz et al., 2013*; *Simon, 2013*; *West et al., 2014*). This technology was based on an affinity purification of Neat1 RNA together with its protein partners and mRNA targets by using oligonucleotides that are complementary to Neat1 RNA sequences. In our adapted technology, anti-sense oligonucleotides were designed in stretches of Neat1 RNA available for hybridization and not occluded by in silico predicted secondary structure (*Figure 4—figure supplement 1*). With this strategy, we found that two anti-sense oligonucleotides allowed a 30 to 40-fold enrichment in Neat1 RNA compared to Neat1 RNA input from cross-linked GH4C1 cellular extracts (*Figure 4—figure supplement 2*). We performed Neat1 RNA pull-down using these two specific biotinylated complementary oligonucleotides that target Neat1 and one biotynylated irrelevant probe in cell lines stably expressing Alu-containing egfp mRNA or IRAlu-containing egfp mRNA. While one of the specific oligonucleotide (S oligo 2) is more efficient than the other (S oligo 1) to pull-down Neat1, the enrichment in Neat1 was not statistically different in the two Alu-egfp and IRAlu-egfp cell lines (*Figure 4E*). By contrast, the amounts of egfp mRNA retrieved after Neat1 RNA pull-down by the two specific probes compared to the non-specific probe were significantly higher in IRAlu-egfp compared to Alu-egfp cell line (*Figure 4F*). This result clearly showed that IRAlu-egfp mRNA was preferentially associated with Neat1-containing paraspeckles compared to Alu-egfp mRNA.

## IRAlu element induced egfp mRNA circadian nuclear retention

Since we showed that paraspeckles displayed a circadian expression in GH4C1 cells, we asked whether IRAlu-egfp mRNA shown here to be associated with paraspeckles could be rhythmically retained in the nucleus. To answer this question, we fractionated cytoplasmic and nuclear RNAs in IRAlu-egfp and Alu-egfp cells harvested every 4 hr during 44 hr. We showed that in addition to the previously described higher ratio of nuclear versus cytoplasmic egfp mRNA in IRAlu-egfp cell line compared to Alu-egfp cell lines (*Figure 4D*) we also found high amplitude circadian variations in this ratio in IRAlu-egfp cell line (*Figure 5A*). Whereas it was observed that the nuclear/cytoplasmic ratio of egfp mRNA could also be fitted by a cosinor equation in Alu-egfp cell line, the magnitude of this rhythmic pattern was highly reduced (*Figure 5A*) as compared to IRAlu-egfp cell line (*Figure 5— source data 1*).

## Loss of circadian nuclear egfp-IRAlu RNA retention after paraspeckle disruption

To address the issue whether the presence of paraspeckles was necessary to the rhythmic egfp mRNA nuclear retention in IRAlu-egfp cell line, we employed two strategies to knockdown Neat1 RNA, namely small interfering RNA (siRNA) and antisens oligonucleotides (ASO). Neat1 has been previously shown by others to be effectively knocked down by siRNA (*Chen and Carmichael, 2009*; *Clemson et al., 2009*), and we indeed verified by RT-qPCR that Neat1 RNA levels were reduced in IRAlu-egfp cell line to around 60% after treatment with specific siRNA compared to negative control siRNA, a decrease that was comparable to that obtained with Neat1 ASO (*Figure 5—figure supplement 1A*); more importantly, whereas Neat1 rhythmic expression pattern was not affected by negative control (*Figure 5—figure supplement 1B*), it was suppressed both by Neat1 siRNA and Neat1 ASO (*Figure 5—figure supplement 1C,D*). In addition, when IRAlu-egfp cells were transfected either by Neat1 siRNA or Neat1 ASO as compared to negative control, the relative ratio of nuclear versus cytoplasmic egfp mRNA levels was significantly decreased (*Figure 5B*). Moreover, whereas the circadian egfp mRNA nuclear retention was not affected after transfection of the cells by the negative control (not shown), it was abolished after both Neat1 siRNA and Neat1 ASO transfection (*Figure 5C*). These loss of function experiments showed that the IRAlu-egfp reporter does not cycle when paraspeckles are disrupted.

## IRAlu element induced EGFP protein circadian expression

After cell synchronization, EGFP fluorescence was recorded with real-time video microscopy. Individual cells included in the analysis are those that can be monitored during at least 48 consecutive hours and whose EGFP fluorescence intensity is 10% higher than the background. Values obtained in these selected cells were fitted by a cosinor equation. Example of data from two cells that could (right part) or could not (left part) be fitted by cosinor equation were given in *Figure 5D*. An example of a rhythmic fluorescence pattern over 48 consecutive hours in an individual cell was also given in *Video 1*. The percent of cells that could be fitted by cosinor equation with a $R^2>0.5$ was then calculated for each cell line. This percent was significantly higher in IRAlu-egfp cell line compared to the Alu-egfp and egfp cell lines (*Figure 5D*) allowing to show that insertion of an IRAlu element in the 3'-UTR of egfp mRNA promoted EGFP protein circadian expression.

## Characterization of paraspeckle associated mRNA

### Analysis of Neat1 RNA pull-down by RNA sequencing

To preserve weak in situ interactions, GH4C1 cells were cross-linked. RNA pull-down was performed using two specific biotinylated oligonucleotides (S oligo 1 and S oligo 2) and a non-specific biotinylated oligonucleotide. Libraries were generated from the purified RNAs obtained with the two specific oligonucleotides but no library could be obtained with the non-specific oligonucleotide due to the too small quantity of material recovered. After sequencing of the two libraries, results were analyzed using the Tophat/Cufflinks pipeline (*Trapnell et al., 2012*). Only transcripts which exhibited values of fragment per kilobase per million of mapped reads (FPKM) higher than 1 were taken into account. We assessed the specificity of Neat 1 RNA pull-down by crossing the FPKM>1-limited lists obtained with the two specific oligonucleotides. 4268 genes (28% of expressed transcripts in GH4C1

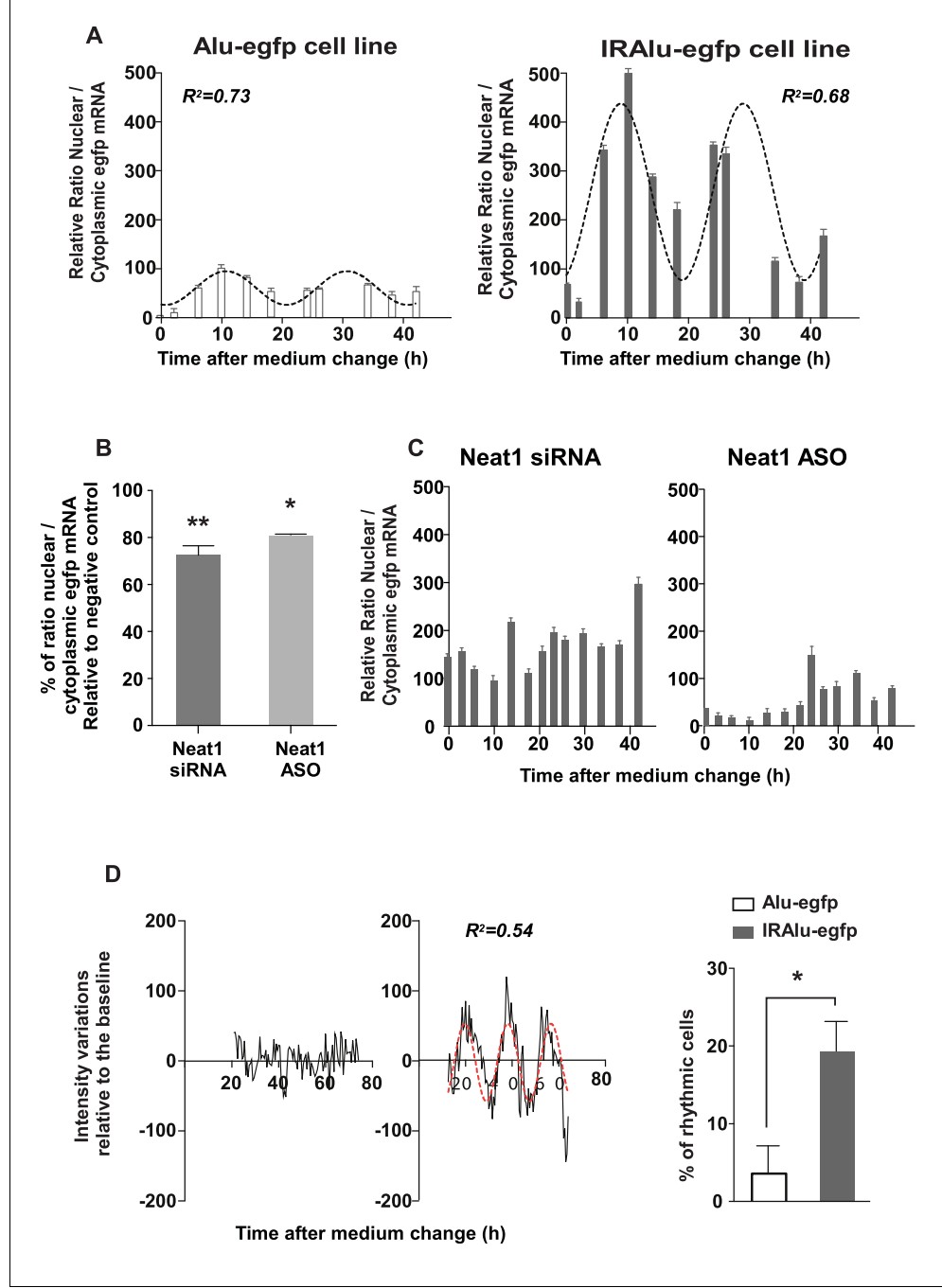

**Figure 5.** IRAlu element induced egfp mRNA circadian nuclear retention and EGFP circadian cytoplasmic expression. (**A**) Rhythmic ratio of nuclear versus cytoplasmic egfp mRNA levels in Alu-egfp and IRAlu-egfp cell lines. Experimental values (n=3 for each cell line) can be adequately fitted ($R^2$>0.55) with a non-linear cosinor equation in which the period value is set to 24 hr (see also *Figure 5—source data 1*). (**B–C**) Effects of paraspeckle disruption by Neat1 siRNA or Neat1 antisens oligonucleotides (ASO) on egfp mRNA nuclear retention in IRAlu-egfp cells. Nuclear and cytoplasmic egfp mRNA were quantified by qPCR in each condition and normalized to the relative amount of gapdh mRNA (n=2 for each condition). (**B**) Ratio of nuclear versus cytoplasmic egfp mRNA levels are compared between negative control and Neat1 siRNA or ASO *p<0.05 **p<0.01 (**C**) Loss of rhythmic egfp mRNA nuclear retention in IRAlu-egfp cells after Neat1 siRNA or ASO. Experimental values (n=2 for each condition) cannot be adequately fitted ($R^2$<0.55) with a non-linear cosinor equation in which the period value is set to 24 hr. (**D**) eGFP fluorescence was recorded with real-time video microscopy in individual cells and values were fitted by a cosinor equation. Shown are examples of data from two cells that could not or could be fitted by

*Figure 5 continued on next page*

*Figure 5 continued*

cosinor equation, in left and right panels respectively. The percent of cells that could be fitted by cosinor equation with a $R^2 > 0.55$ was then calculated for each cell line. This percent was significantly higher in IRAlu-egfp cell line compared to the Alu-egfp and egfp cell lines. *p<0.05

The following source data and figure supplement are available for figure 5:

**Source data 1.** Cosinor analysis of the rhythmic ratio of nuclear versus cytoplasmic egfp mRNA levels in Alu-egfp and IRAlu-egfp cell lines.

**Figure supplement 1.** Effects of knockdown of Neat1 by Neat1 siRNA or Neat1 ASO.

cells) were common to the two lists (*Figure 6—source data 1*). This represented 65% of the list obtained with S oligo 1 and 83% of the list obtained with S oligo 2.

## Association of rhythmic transcripts with paraspeckles

Using publicly available datasets, we then tried to evaluate whether known cycling transcripts can be shown to be associated with paraspeckles. To allow comparison between gene lists generated from different species, namely rat and mouse, we used only official gene symbols (3928 genes in our 4268 gene list if we consider only official gene symbols). We first compared the mouse pituitary circadian transcriptome previously published (*Hughes et al., 2007*) with our data set of Neat1 RNA targets. It appeared that 68 genes out of 362 from the list of pituitary circadian transcripts (18.8% ) were found to be included in our data set (*Figure 6A*). We then tried to assess whether genes known to be post-transcriptional cyclers are associated with paraspeckles. To this end, we compared the list of putative post-transcriptional cycling transcripts established by Menet et al. (*Menet et al., 2012*) in the mouse liver with our list of Neat1 RNA targets. Although these two lists have been established in two different tissues from two different species, it appeared that 182 genes out of 675 from the list of post-transcriptional circadian genes in the liver (27%) were found to be included in our data set (*Figure 6A*). In agreement with the assumption that rhythmic circadian genes are tissue specific, we found very few genes (24 genes) common to the list of Hughes et al. (<7%) (*Hughes et al., 2007*) and that of Menet et al. (<4%) (*Menet et al., 2012*). In spite of this reduced overlap, we found that 6 genes out of these 24 common genes (25% ) were Neat1 RNA targets (*Figure 6A*).

## Circadian nuclear retention of Neat1 RNA target genes

We then tested on a few genes whether Neat1 RNA targets as identified here can display a circadian nuclear retention. With this aim, we chose among Neat1 RNA targets, six genes that were included either in the list of Menet et al. (*Menet et al., 2012*) namely Calr, Evi5 and Maged1 or in the list of Hughes et al. (*Hughes et al., 2007*) namely Canx, Pcbp2 and Syne1 and one gene that was common to both lists namely Fkbp4 (*Figure 6A*). First we verified by qPCR the specific association of the seven selected genes with Neat1. To this end, we determined the enrichment in these seven mRNA obtained after Neat1 RNA pull-down with the two specific biotinylated complementary oligonucleotides that target Neat1 compared to the biotinylated irrelevant probe. As shown in *Figure 6—figure supplement 1*, the seven selected genes were significantly enriched after Neat1 RNA pull-down by the two specific probes compared to the non-specific probe. We further found that the seven selected genes displayed a rhythmic pattern of nuclear retention in IRAlu-egfp cells since values obtained for their relative mRNA ratio between nucleoplasm and cytoplasm over the time can be fitted by a sine wave curve with a R squared >0.55 (*Figure 6B*, *Figure 6—figure supplement 2*, *Figure 6—source data 2*). It may be noticed that the minimum value of the relative ratio nuclear / cytoplasmic mRNA occurred roughly 20 hr after the medium

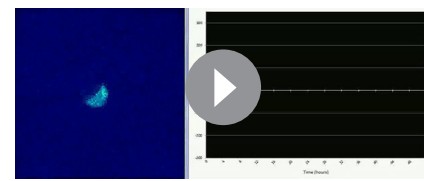

**Video 1.** Rhythmic fluorescence pattern of EGFP expression over 48 consecutive hours in an individual cell from IRAlu-egfp cell line.

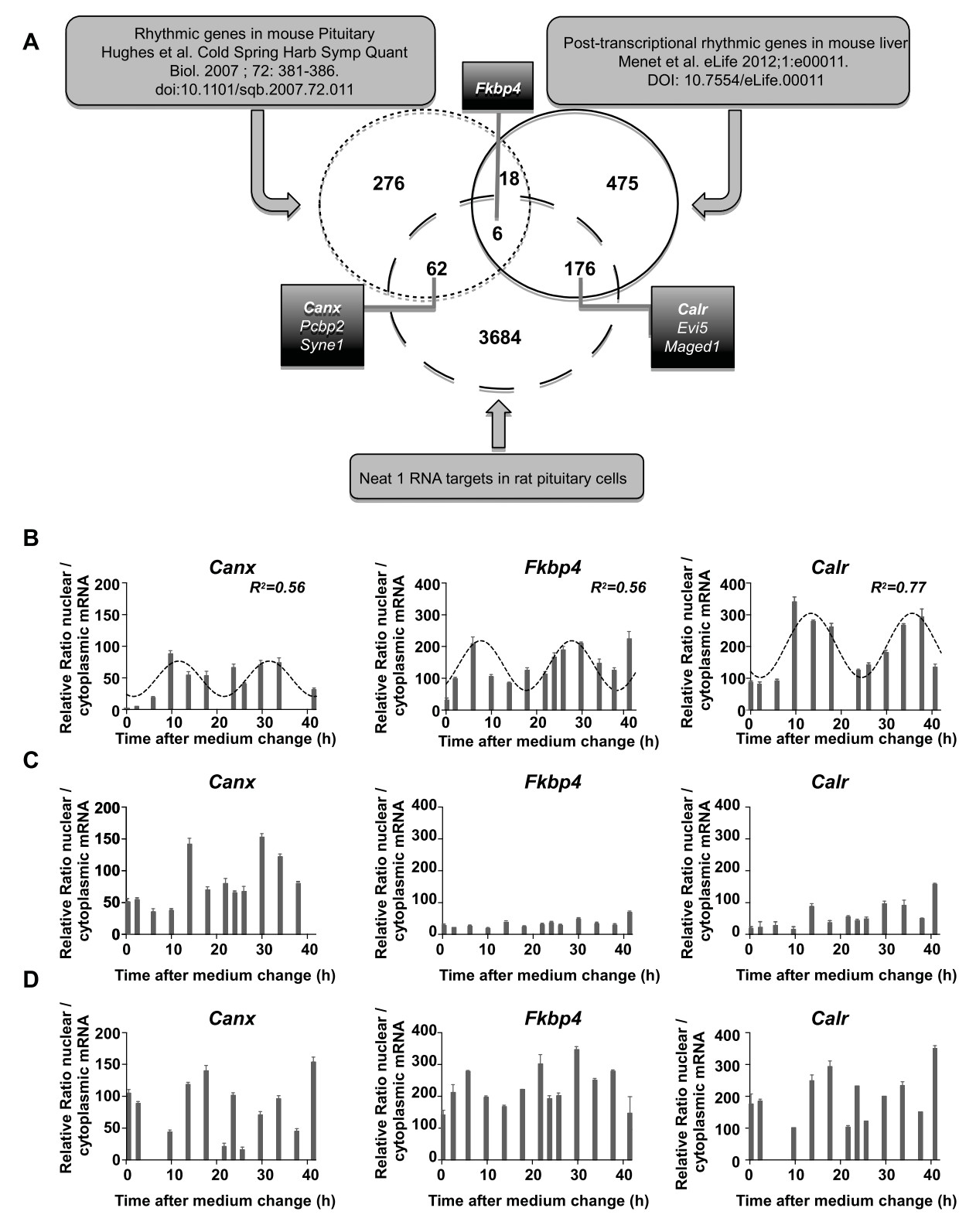

**Figure 6.** Characterization of paraspeckle-associated mRNA. (**A**) Venn diagram representation of the overlap between transcripts linked to the paraspeckle (Neat1 RNA targets) with transcripts known as rhythmic genes in the mouse pituitary gland (*Hughes et al., 2007*) and/or post-transcriptional rhythmic genes in mouse liver (*Menet et al., 2012*). To allow comparison between lists from two different species, lists of genes were restricted to official gene symbol lists. Shown are the seven genes that were selected for analysis of mRNA nuclear/cytoplasmic ratio; in bold the three

*Figure 6 continued on next page*

*Figure 6 continued*

genes illustrated in the figure. (B) Rhythmic ratio of nuclear versus cytoplasmic mRNA levels of three selected genes. Experimental values (n=2) can be adequately fitted ($R^2>0.55$) with a non-linear cosinor equation in which the period value is set to 24 hr (see also *Figure 6—source data 1*). (C–D) Loss of rhythmic ratio of nuclear versus cytoplasmic mRNA levels of 3 selected genes after Neat1 siRNA (C) or Neat1 ASO (D). Experimental values (n=2) cannot be adequately fitted ($R^2<0.55$) with a non-linear cosinor equation in which the period value is set to 24 hr.

The following source data and figure supplements are available for figure 6:

**Source data 1.** List of Neat1 RNA target genes.

**Source data 2.** Cosinor analysis of the relative ratio nuclear/cytoplasmic mRNA of seven Neat1 RNA targets in GH4C1 cells.

**Figure supplement 1.** Enrichment relative to input in seven selected mRNA after Neat1 RNA pull-down.

**Figure supplement 2.** Rhythmic ratio of nuclear versus cytoplasmic mRNA levels of four NEAT1 RNA target genes.

change, i.e. at a time when the levels of paraspeckle protein components and Neat1 RNA were the lowest (*Figure 3*).

## Loss of circadian nuclear retention of Neat1 RNA targets after paraspeckle disruption

To determine whether mRNA retention in paraspeckles can drive the oscillations of selected transcripts, we performed a loss of function experiment. In *Figure 6C–D* and *Figure 6—figure supplement 2*, it was shown that knockdown of Neat1 RNA, either by Neat1 siRNA (*Figure 6C*) or by Neat1 ASO (*Figure 6D*) disrupted the circadian pattern of the ratio of nuclear versus cytoplasmic levels of the seven selected mRNA as assessed by the inability to fit the values obtained with a sine wave curve with a R squared >0.55. These results showed that paraspeckles play a role in the rhythmic expression of theses selected genes.

## Discussion

### Features of paraspeckle bodies in GH4C1 cells

Paraspeckles are found in almost all of the cultured cell lines and primary cultures from tissues (*Bond and Fox, 2009*), except for embryonic stem cells (*Chen and Carmichael, 2009*). Four RNA-binding proteins, including three members of the Drosophila melanogaster behavior human splicing (DBHS) family proteins (NONO, PSPC1 and SFPQ) and RNA-binding motif protein 14 (RBM14) are classified as 'classical' paraspeckle protein components (*Bond and Fox, 2009*; *Prasanth et al., 2005*; *Fox et al., 2002*; *Nakagawa and Hirose, 2012*). Both endogenous and tagged forms of these proteins are found localized within the nucleoplasm as well as paraspeckles in mammalian cells. With a confocal fluorescent study, we report in a rat pituitary cell line that all four paraspeckle protein components are distributed both in the nucleoplasm and in subnuclear foci. Among these later, we identify paraspeckle nuclear bodies by overlapping side-by-side the staining of two paraspeckle protein components. This allows not only to provide evidence for the existence of paraspeckles in rat pituitary cell nuclei but also to show that as already reported in other cell types, paraspeckles are small, irregularly sized and unevenly distributed subnuclear bodies. Using RIP experiments, we further show that all four paraspeckle protein components are associated with the specific paraspeckle localized lncRNA Neat1. This later is further detected by RNA FISH. Confirming previous reports in other cell types, when identified by a combination of Neat1 RNA FISH and confocal microscopy, paraspeckles appear as round foci in rat pituitary cells. The diameter of such round foci is in the range of that estimated in previous studies (around 360 nm). However after use of a combination of Neat1 RNA FISH and STORM analysis, paraspeckles appear more likely as oblong structures with smaller dimensions reminiscent of the interchromatin granule-associated zone (IGAZ) that had been observed under electron microscope and had been reported to correspond to paraspeckles (*Cardinale et al., 2007*; *Bond and Fox, 2009*; *Souquere et al., 2010*). Width and height

measurements obtained after STORM analysis of Neat1 RNA FISH staining are smaller than the dimensions of IGAZ nuclear bodies established by an electron microscopy analysis in human and mouse cells (*Souquere et al., 2010*) suggesting that either IGAZ/paraspeckle nuclear bodies are smaller in rat than in mouse cells or alternatively and more likely, Neat1 RNA staining does not fill in all the nuclear bodies as already proposed (*Souquere et al., 2010*).

In a previous study, we reported that two paraspeckle proteins, namely NONO and SFPQ, display a circadian expression pattern in rat pituitary cells (*Guillaumond et al., 2011*) and we find here that it is also the case for the two other 'classical' paraspeckle proteins, PSPC1 and RBM14. Furthermore, all four proteins bind rhythmically to Neat1, suggesting that the pools of proteins involved in paraspeckle composition vary with time. In addition, the lncRNA Neat1 itself displays a circadian expression pattern and we further found using a pair of primers allowing the specific detection of the long form of Neat1 RNA, Neat1-2 (*Figure 2—source data 1*) that the expression of this form known to be sufficient for the formation of paraspeckle displays a circadian pattern (data not shown). The circadian expression pattern of Neat1 RNA is not the privilege of 'in vitro' pituitary cells since it is also found 'ex vivo' in several mouse tissues including the pituitary gland but also numerous other peripheral oscillators such as the spleen or the adrenal gland. This holds also true for the central clock, namely the suprachiasmatic nuclei. Finally, in GH4C1 cells, the circadian variations in Neat1 RNA levels are shown here to reflect rhythmic variations in paraspeckle number within the cells. It may then be assumed that the circadian expression pattern of Neat1 RNA described here in a number of mouse tissue also reflects variations in paraspeckle number although it cannot be excluded that paraspeckle size could also oscillate over the circadian period both in pituitary cells and in other oscillators.

## Association of IRAlu elements with paraspeckles

As previously reported by Chen and Carmichael (*Chen et al., 2008*), we found that a pair of IRAlus in the 3'-UTR of egfp can strongly silence EGFP expression in GH4C1 pituitary cell line and after fractionating cytoplasmic and nuclear RNAs from our two cell lines, we observed that IRAlus-containing egfp mRNA appear to be preferentially retained in the nucleus in comparison with those having a single Alu element. It then appeared that silencing EGFP expression mechanism relied on egfp mRNA nuclear retention.

To obtain evidence that nuclear retained IRAlus are associated with paraspeckle bodies, we immunoprecipitated RNA complexes containing paraspeckle protein PSPC1 and characterized the associated egfp mRNA. We found IRAlu-containing egfp mRNA to be associated with PSPC1 whereas Alu-containing egfp mRNA was not. While PSPC1 as well as NONO and SFPQ accumulate in paraspeckles, these proteins are also found elsewhere throughout the nucleoplasm and thus may have functions apart from paraspeckle retention. In contrast to paraspeckle-associated proteins that are multifunctional, NEAT1 may have only one function. We then developed a technique to characterize nuclear mRNA Neat1 targets using a Neat1 RNA pull-down approach that allowed us to show the association of IRAlu egfp mRNA with Neat1 providing a much more convincing evidence for a nuclear retention of IRAlu egfp mRNA by paraspeckles.

## IRAlu element-induced circadian expression of reporter gene

Given their presumed functions in gene expression through corresponding mRNA nuclear retention and since they display a circadian expression pattern, it may be assumed that paraspeckle bodies can rhythmically retain RNAs in the nucleus leading to a rhythmic expression of the corresponding gene. In keeping with this view, we showed that IRAlu elements inserted in 3'-UTR of egfp reporter mRNA allow for its circadian retention within the nucleus and consequently for EGFP protein circadian expression. However, it may be noticed that egfp mRNA displayed also a circadian nuclear retention when a single antisens Alu element was inserted in its 3'-UTR but in this case the variations of the ratio over the circadian period was highly reduced in contrast to the high amplitude circadian variations of the nuclear versus cytoplasmic ratio observed in IRAlu-egfp cell line. These small amplitude variations of the ratio between nuclear and cytoplasmic mRNA contents in Alu-egfp cell line were further accompanied by a weak number of cells expressing a circadian pattern of EGFP expression. Since antisens Alu elements within the 3'-UTR of mRNAs can be the target of several RNA-binding proteins (*Kelley et al., 2014*; *Zarnack et al., 2013*) it may be speculated that some of these

RNA-binding proteins display a rhythmic expression pattern accounting for the weak EGFP circadian expression observed in Alu-egfp cell line. Among the identified RNA-binding proteins that target antisens Alu elements within the 3'-UTR of mRNAs, HNRNPC was shown to display a circadian expression pattern in several tissues including the mouse pituitary (see CIRCA: http://circadb.hoge-neschlab.org/). This is not the case for HNRNPU and STAUFEN 1 (STAU1)-mediated messenger RNA decay we did not find to display a circadian expression pattern in GH4C1 cells (*Guillaumond et al., 2011*) and (data not shown) and are then unlikely involved in circadian EGFP expression in Alu-egfp cell line. The mechanism involved in the circadian nuclear retention of Alu-containing egfp mRNA remains anyway to be determined.

## Contribution of paraspeckle nuclear retention to circadian gene expression

Nuclear retention of egfp mRNA in IRAlu-egfp cells is no more cycling when paraspeckles were disrupted assuming that the nuclear retention of the reporter gene by paraspeckles contributes to cycling. To determine whether this paraspeckle impact on rhythmic gene expression occurs 'in vivo' on a much larger scale, we characterized in our pituitary cells the list of the putative Neat1 RNA target genes and we analyzed in publicly available datasets the known cycling transcripts that are included in this Neat1 RNA target list. Whereas it is well known that rhythmic circadian genes are tissue specific and that it is unfavorable to compare datasets obtained in two different species we found a robust overlap of our gene list with circadian genes in the mouse pituitary (*Hughes et al., 2007*) and a closer overlap with post-transcriptional circadian genes in the mouse liver (*Menet et al., 2012*). Furthermore, on a few circadian genes common to either of these lists or both, we showed a loss of rhythmic pattern when paraspeckles were disrupted assuming that paraspeckle retention plays a relevant role in circadian gene expression. It is now widely admitted that post-transcriptional circadian regulation plays a major role in determining oscillations at the mRNA level. In agreement with this view, the new mechanism described here suggests that circadian mRNA oscillations could be post-transcriptionally controlled through rhythmic nuclear retention by paraspeckle nuclear bodies.

# Materials and methods

## Cell line culture and preparation of stably transfected cell lines

GH4C1 cells, a rat pituitary somatolactotroph line, were obtained in 2012 from ATCC (CCL-82.2, lot number: 58945448, Molsheim, France) with certificate analysis and were confirmed to be free of mycoplasma (MycoAlert, Lonza, Levallois-Perret, France). They were grown in HamF10 medium supplemented with 15% horse serum and 2% fetal calf serum. To synchronize cells between themselves, GH4C1 cells were transferred to fresh medium and were harvested after this fresh medium replacement (T0) every 4 hr from T2 to T30-38. For the generation of stable GH4C1 cell lines, cells were transfected with the plasmid constructs expressing a neomycin resistance gene by Lipofectamine Plus (Invitrogen, Cergy Pontoise, F). Cells were selected with 250 µg/ml G418 (Invitrogen) beginning 48 hr after transfection.

## Plasmid constructs

Alu and IRAlus elements from Nicn1 were cloned by PCR from genomic DNA obtained from Hela cells. Sequences were then inserted in the pEGFP-C1 (Clontech, Mountain View, CA) at the BglII and HindIII sites as described by Chen et al. (*Chen and Carmichael, 2008*).

## Knockdown of Neat1

Predesigned siRNAs targeting rat Neat1 were purchased from GeneCust (siRNA Set, GeneCust, Dudelange, Luxembourg). IRAlu-egfp cells plated in 6-well dishes were transfected with 500 pmol of a pool of four siRNA (125 pmol each, see *Figure 2—source data 1* for the sequences) using Lipofectamine 2000 (Life Technologies, Saint Aubin, France). As controls, 500 pmol of non-targeting siRNA from GeneCust (negative control siRNA, see *Figure 2—source data 1* for the sequences) were used. Cells were harvested 48 hr after transfection.

The antisense oligonucleotides (ASO) used for Neat1 knockdown experiments were phosphoro-thioate modified at their backbone to increase their stability. Two Neat1 ASO were pooled that were targeted to regions common to NEAT1_1 and NEAT1_2 isoforms (see *Figure 2—source data 1* for the sequences). IRAlu-egfp cells plated in 6-well dishes were transfected with 300 pmol of the pool of the two ASO (150 pmol each, see *Figure 2—source data 1* for the sequences) using Lipo-fectamine 2000 (Life Technologies). Cells were harvested 48 hr after transfection.

## RNA expression analysis

Nuclear, cytoplasmic or total RNA was prepared from GH4C1 cells using an RNA XS kit (Macherey Nagel, Hoerdt, France). Total RNA purification was performed on 24-well cell dishes. Nuclear and cytoplasmic RNA isolation was performed using 10 cm cell dishes that were rinsed twice with ice-cold PBS, incubated in 1 ml of ice-cold cell lysis buffer A (10 mmol/L Tris pH 7.4, 3 mmol/L MgCl2, 10 mmol/L NaCl and 0.5% NP-40). Nuclei and cytoplasma were separated by centrifugation (500 x *g* for 5 min). One-sixth of the supernatant was used to prepare cytoplasmic RNA. To obtain pure nuclear RNA, the nuclear pellets were subjected to two additional washes with 1 ml lysis buffer A and were then extracted with XS kit reagent. Total RNA (500 ng) was then used for cDNA synthesis performed with a High Capacity RNA to cDNA kit (Applied Biosystem, Courtaboeuf, France). Real-time PCR was performed on a 7500 fast Real-Time qPCR system (Applied Biosystems) using Fast SYBR Green mix (Applied Biosystems). The sequences of the primers used in qPCR are given in *Figure 2—source data 1*. mRNA accumulation was normalized relative to Gapdh mRNA levels.

## Western-blot analysis

Total or nuclear protein extracts prepared as previously described (*Becquet et al., 2001*) from confluent GH4C1 cells grown in 10 cm dishes, were submitted to Western-blot analysis as previously described (*Guillaumond et al., 2011*) with polyclonal primary antibodies raised against PSPC1 (1:1000, sc-84576, Santa Cruz Biotechnology, Heidelberg, Germany), RBM14 (1 µg/ml, Abcam, Paris, France), ATF2 (1/1000, sc-187, Santa Cruz Biotechnology), GFP (1/1000, A6455, Molecular Probes, Paisley, UK) and monoclonal primary antibody raised against α-Tubulin (1/1000, T 6199, Sigma-Aldrich, Saint-Louis, USA). The ratio of PSPC1 and RBM14 to ATF2, a constitutive transcription factor, and the ratio of GFP to tubulin were determined by densitometry using ImageJ software (National Institutes of Health, USA).

## RNA immunoprecipitation (RIP) experiments

GH4C1 cells grown in 10 cm dishes were rinsed two times with 5 ml cold phosphate buffer saline (PBS). Cells were then harvested by scraping in ice-cold PBS and transferred to a centrifuge tube. After centrifugation (2500 x *g* for 5 min) cells were pelleted and suspended in 100 µl of Polysome lysis buffer (PLB; 10 mM HEPES, pH 7.0, 0.1M KCl, 5 mM MgCl2, 25 mM EDTA, 0.5% NP40, 1 mM DTT, 100 U/ml RNAse OUT and complete protease inhibitor cocktail). After mixing by pipetting up and down, cells were kept on ice for 5 min to allow the hypotonic PLB buffer to swell the cells. The cell lysate was then aliquoted and stored at −80°C.

Cell lysate was centrifuged at 14,000 x *g* for 10 min at 4°C and diluted 1/100 in NET2 buffer (NET2 buffer corresponded to NT2 buffer: 50 mM Tris-HCl, pH 7.4, 150 mM NaCl, 1 mM MgCl2 and 0.05% NP40 added with 1 mM DTT, 20 mM EDTA, 200 U/ml RNAse Out). An aliquot of diluted cell lysate was removed and represented the starting material or 'input' which was processed alongside the immunoprecipitation to compare with immunoprecipitated mRNAs at the end. RIP experiments were performed overnight at 4°C on diluted cell lysate with antibodies to NONO (ab45359, Abcam), SFPQ (ab38148, Abcam), PSPC1 (SAB4200068, Sigma-Aldrich, Saint-Quentin Fallavier, France) and RBM14 (ab70636, Abcam) or non-specific rabbit polyclonal antibody (anti-Furin, sc-20801, Santa-Cruz Biotechnology). After incubation was completed, 15 µl of Magna ChIP protein A magnetic beads (16–661, Millipore, Molsheim, France) were added for 1h at 4°C. Beads were washed 6 times with cold NT2 buffer and treated by proteinase K for 30 min at 55°C. RNA eluted from beads was purified using Nucleospin RNA XS (Macherey-Nagel) and processed for cDNA synthesis using a High Capacity RNA to cDNA kit (Applied Biosystems).

## Neat1 RNA pull-down

Neat1 RNA pull-down is a hybridization-based strategy that uses complementary oligonucleotides to purify Neat1 RNA together with its targets from reversibly cross-linked extracts. In cross-linked extracts, it is expected that some regions of the RNA will be more accessible for hybridization than others due in particular to secondary structure. To design oligonucleotides that target these regions and then can hybridize specifically to Neat1 RNA, we modeled the secondary structure of Neat1 RNA using bioinformatics (MaxExpect software [*Lu et al., 2009*]). Two antisense DNA oligonucleotide probes that target accessible regions of the lncRNA Neat1were designed and used for Neat1 RNA specific pull-down (*Figure 4—figure supplement 1*). One biotynylated irrelevant probe was used for Neat1 RNA non-specific pull-down. All three probes were biotinylated at the 3′ end (*Figure 2—source data 1*).

Briefly, 10 cm cell dishes were incubated in 1 ml of ice-cold cell lysis buffer A as described above. Nuclei were scraped and separated by centrifugation (500 x *g* for 5 min). The nuclear pellets were then fixed with 1% paraformaldehyde in PBS for 10 min at room temperature. Crosslinking was then quenched with 1.25 M glycine for 5 min. Cross-linked nuclei were rinsed two times again with PBS and pelleted at 500 x *g*. Nuclear pellets were stored at −80°C. To prepare lysates, nuclear pellets were suspended in lysis buffer (50 mM Tris, pH 7.0, 10 mM EDTA, 1% SDS added with a protease inhibitor cocktail and RNAse-Out), and sonicated using BioruptorPlus (Diagenode, Seraing , Belgium) by 2 pulses of 30 s allowing complete lysate solubilization. RNA was in the size range of 400 to 2000 nucleotides. Nuclear lysates were diluted V/V in hybridization buffer (750 mM NaCl, 1% SDS, 50 mM Tris, pH 7.0, 1 mM EDTA, 15% formamide). The two specific or the non-specific probes (100 pmol) were added to 1.2 ml of diluted lysate, which was mixed by end-to-end rotation at room temperature 4 to 6 hr. Streptavidin-magnetic C1 beads (Dynabeads My OneStreptavidin C1– Invitrogen Life Technologies) were added to hybridization reaction (50 µl per 100 pmol of probes) and the whole reaction was mixed overnight at room temperature. Beads–biotin-probes–RNA adducts were captured by magnets (Invitrogen) and washed five times with a wash buffer (2×SSC, 0.5% SDS). After the last wash, buffer was removed carefully. For RNA elution, beads were suspended in 100 µl RNA proteinase K buffer (100 mM NaCl, 10 mM Tris, pH 7.0, 1 mM EDTA, 0.5% SDS) and 100 µg proteinase K (Ambion). After incubation at 45°C for 45 min, RNA was isolated using NucleoSpinRNA XS (Macherey-Nagel). Eluted RNA was subject to RNA-sequencing or RT–qPCR for the detection of enriched transcripts.

## RNA sequencing

The construction of Illumina DNA libraries from RNA pools obtained with each of the two specific oligonucleotides and the sequencing were performed by Helixio (Clermont-Ferrand, France). Libraries were prepared with Illumina TruSeq Stranded mRNA Sample Preparation kit. Sequencing was done on Illumina NextSeq 500 (60 million reads per sample on average).

Analyses were performed on a local instance of Galaxy (*Afgan et al., 2016*). Paired-end reads (75bp) were aligned to the Rat reference genome (Rnor_6.0.80, Ensembl) using TopHat2 v2.209 with default parameters (*Kim et al., 2013*). We further used Cuffflinks v2.2.1 (*Trapnell et al., 2012*) to quantify mRNA level and fragment per kilobase per million of mapped reads (FPKM) were calculated. Consistent with prior studies (*Nagaraj et al., 2011*; *Hart et al., 2013*), the log2 (FPKM) distribution shows a primary peak of high expression genes, with a long left shoulder of low-expression transcripts. Therefore, only transcripts with FPKM values higher than 1 were taken into account. In order to assess the specificity of the NEAT1 RNA pull-down we crossed the results obtain with the two oligonucleotides (see Neat1 RNA pull-down) to generate the list of transcripts associated to the paraspeckles.

All the RNA sequencing data are available at Gene Expression Omnibus (GEO) (accession no. GSE81972).

## Multiple-label fluorescence immunocytochemistry

GH4C1 cells were grown on glass coverslips coated with poly-ornithine. After incubation in 30% normal blocking serum containing 0.2% Triton X-100 (30 min), coverslips were transferred to primary antibody or a mixture of primary antibodies (overnight, 4°C) and, after rinsing, to fluorescent secondary antibody or a mixture of fluorescent secondary antibodies (2 hr, 22°C). Nuclei were stained with

Hoechst dye (1 µmol/ml, 5 min). All dilutions and rinsing between each step of the procedures were made in Tris buffer added with 1% normal serum.

NONO, SFPQ, PSPC1, RBM14 were detected using the respective polyclonal rabbit antibodies: ab45359 (1: 1000) Abcam, ab38148 (1: 1000) Abcam, HPA038904 (1: 1000) Sigma-Aldrich, ab70636 (1: 1000) Abcam.

After mounting with PBS containing 50% glycerol, the reacted sections were kept in darkness at 4°C until examination.

## RNA-FISH

To detect Neat1 RNA, GH4C1 cells grown on glass coverslips coated with poly-ornithine were fixed in 3.6% formaldehyde. Hybridization was carried out using Custom Stellaris FISH Probes (Biosearch Technologies, Novato, USA). Probes used are Neat1 probes labeled with Quasar 570 Dye or Neat1 probes labeled with Quasar 670 for STORM analysis. Nuclei of the cells were counterstained by Hoechst solution (5 ng/ml).

## Confocal microscopy analysis

The confocal image acquisition was performed on a Zeiss LSM780 confocal microscope equipped with a 63X 1.4 N.A. oil immersion objective. Dyes used in either immunocytochemistry (ICC) or RNA-Fish were imaged using the appropriate wavelength for optimal dye excitation, i.e. 405 nm for Hoechst, 561 nm for Cy3 and Quasar 570, and 633 nm for both Cy5 and Quasar 670. The spectral detection of emitted fluorescence were set as follows: 420–480 nm for Hoechst, 560–670 nm for Cy3 or Quasar 570 and 640–750 nm for Cy5 or Quasar 670. Three dimensional z-stacks were collected automatically as frame by frame sequential image series. To enhance resolution, the ICC images were deconvolved based on a theoretical point spread function (PSF) using AutoQuantX (Media Cybernetics, Rockville, USA).

## STORM microscopy

Prior to observation, samples were mounted in imaging buffer (50 mM Tris pH 8.0, 20 mM NaCl, 0.5 mg/ml glucose oxydase, 40 µg/ml catalase, 10% w/v glucose, 10 mM mercaptoethylamine). Imaging was performed with a Nikon Storm System (N-Storm, Nikon France S.A, Champigny sur Marne, France) equipped with a 100X 1.49 N.A. oil immersion objective. A reference conventional fluorescence image of the cell of interest was acquired before recording storm sequence of images. For super resolution image acquisition, the sample stained with Quasar 670 dye was excited by a 647 nm imaging laser. Illumination with 405 nm light was gradually increased along recording to reactivate Quasar 670 blinking. The fluorescence emission was separated from the excitation light by adapted dichroic mirror (660 nm) and filters (692/4 nm) and collected on an IXon EMCCD camera (Andor Technology, Belfast, UK). Typically, dSTORM images were reconstructed from 40000 frames using NIS-software (Nikon).

## Real-time monitoring of Egfp fluorescence by video-microscopy and mathematical analysis

For real-time monitoring of intracellular fluorescence circadian oscillations, $4 \times 10^4$ Alu-egfp or IRAlu-egfp GH4C1 cells were seeded on glass-bottom 24 well plates (coated with poly- ornithine). The next day, the medium was changed with a growth medium containing 100 nM forskolin to synchronize the cellular circadian oscillators. 20 min later, the medium was changed again with a DMEM without phenol red medium containing 100 mM 4-(2-hydroxyethyl)-1-piperazineethanesulfonic acid (HEPES) and 100 mM pyruvate. The next day, plates were transferred to an inverted microscope (Axiovert-200, Zeiss) equipped with an environmental chamber, the temperature of which was set to 37°C and $CO_2$ regulated to 5%. Egfp was excited through a 475/40 band-pass filter. An EMCCD camera (Rolera EM-C2, Q-Imaging) was used to collect emitted fluorescence at 530/50 nm. Time-lapse recordings of multiple regions in each well were realized over 96 hr with 1 picture every 30 min (Metamorph, Roper Scientific), using a 40X objective. The data were processed for tracking and measuring fluorescence fluctuations of individual cells with the CGE (Circadian Gene Expression) Plugging of ImageJ (*Sage et al., 2010*). Cells retained for analysis fulfilled two criteria: they were

detected for at least 48 hr and their mean fluorescent level was 10% higher than mean level of the background.

## Paraspeckle number quantification

After processing of the cells for FISH of Neat1 RNA, quantification of paraspeckle bodies was performed on confocal images using a 40X objective. At each time point analyzed, 20 to 35 fields acquired in four wells from two different experiments were counted to estimate the total paraspeckle number per well of 100000 cells and to estimate among cells which expressed paraspeckles the mean number per cell.

## Cosinor and statistical analysis

Data from real time monitored cells or mean experimental values ( ± SEM) expressed as a percent of initial value were fitted (Prism4 software, GraphPad Software, Inc.) by a non-linear sine wave equation to find the set of parameters that gives the least-squared distance between the data and the equation: $y(t)= B + A*exp(-t)*sin(2*pi*t/T + P)$ where $B$ is the Baseline, $A$ is the Amplitude, $T$ is the period and $P$ is the Phase-shift. Goodness-of-fit was quantified using R squared, experimental values being considered well fitted by cosinor regression when the R squared was higher than 0.55. A statistically significant circadian oscillation was considered if the 95% confidence interval for the amplitude did not include the zero value (zero-amplitude test) (*Yang et al., 2009*) (*Kavcic et al., 2011*).

Significant differences between groups were determined using either ordinary one-way ANOVA or unpaired t test (Prism4 software). Values were considered significantly different for $p<0.05$ (*).

## Acknowledgements

The authors gratefully acknowledge Nikon France SA (Champigny sur Marne, France) to provide a freely available Nikon Storm System.

## Additional information

### Funding

| Funder | Grant reference number | Author |
|---|---|---|
| Pfizer | SFE award - Research price 2014 | Anne-Marie François-Bellan |

The funders had no role in study design, data collection and interpretation, or the decision to submit the work for publication.

### Author contributions

MT, DB, M-PB, Performed experiments, Analyzed data and interpreted results of experiments, Prepared figures, Edited and revised manuscript; SG, Performed cell culture, Established stably transfected cell lines, Performed plasmid constructs, Analysis and interpretation of data; BB, Performed cell cultures, Transfection experiments, western blot analysis and qPCR analysis; MM, Established stably transfected cell lines and performed FISH experiments, Analysis and interpretation of data; J-LF, Conception and design of research, Performed experiments, Analyzed data and interpreted results of experiments, Prepared figures, Edited and revised manuscript; A-MF-B, Conception and design of research, Performed experiments, Analyzed data and interpreted results of experiments, Prepared figures, Drafted manuscript, Edited and revised manuscript

### Author ORCIDs

Jean-Louis Franc, http://orcid.org/0000-0002-2900-5468
Anne-Marie François-Bellan, http://orcid.org/0000-0002-3278-4642

# Additional files

## Major datasets

The following dataset was generated:

| Author(s) | Year | Dataset title | Dataset URL | Database, license, and accessibility information |
|---|---|---|---|---|
| Torres M, Becquet D, Blanchard M, Guillen S, Boyer B, Moreno M, Franc J, François-Bellan A | 2016 | Determination of the RNA linked to the paraspeckles in the GH4C1 cell line | http://www.ncbi.nlm.nih.gov/geo/query/acc.cgi?acc=GSE81972 | Publicly available at the NCBI Gene Expression Omnibus (accession no: GSE81972) |

The following previously published dataset was used:

| Author(s) | Year | Dataset title | Dataset URL | Database, license, and accessibility information |
|---|---|---|---|---|
| Menet JS, Rodriguez J, Abruzzi KC, Rosbash M | 2012 | Nascent-Seq Reveals Novel Features of Mouse Circadian Transcriptional Regulation | http://www.ncbi.nlm.nih.gov/geo/query/acc.cgi?acc=GSE36916 | Publicly available at the NCBI Gene Expression Omnibus (accession no: GSE36916) |

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
