## [Decision Letter]

Thank you for submitting your article "Circadian RNA expression elicited by 3'-UTR IRAlu-paraspeckle associated elements" for consideration by *eLife*. Your article has been reviewed by three peer reviewers, one of whom is a member of our Board of Reviewing Editors, and the evaluation has been overseen by James Manley as the Senior Editor.

The reviewers have discussed the reviews with one another and the Reviewing Editor has drafted this decision to help you prepare a revised submission.

Summary:

Much of the paper confirms previous results and establishes the role of paraspeckles in rat pituitary cells. The striking results of the paper are the last two figures showing that the levels of the GFP signal from the IRAlu-containing GFP reporter oscillate. This is an important experiment because it directly tests whether the retention of mRNAs in paraspeckles can drive oscillating protein expression. Additional experiments need to be done before this manuscript is acceptable for publication (see below).

Essential revisions:

A loss of function experiment to show that the IRAlu-GFP reporter does not cycle when paraspeckles are disrupted.

A genome wide approach should be used to determine whether known cycling transcripts are associated with paraspeckles.

Test a few additional genes in the in vitro system by comparing RNA levels between nucleoplasm and cytoplasm.

Reviewer #1:

This work posits that the paraspeckles are responsible for post transcriptional control of cyclical retention of mRNAs in the nucleus. They measure the cycling proteins associated with the paraspeckles including Neat1, a long non-coding RNA. Importantly they show that if they place sequences from the inverted Alu repeats in the 3'UTR of an mRNA they don't cycle, they can show both it and its protein product fused to GFP now are nuclear retained and are associated with the paraspeckles.

While this is not my field, it seems that this work takes a step forward in understanding the mechanism of circadian control of some mRNAs, and that the paraspeckles play a role in this process. It has been shown by others cited in their references that parapeckles have been implicated in this process, but this work appears to make a more mechanistic connection. I don't think the STORM imaging adds much to the work, except to refine the resolution of association with the components, but with no additional insights.

Nonetheless, I feel that this work is appropriate for *eLife*.

Reviewer #2:

In this manuscript, the authors investigate the role of paraspeckles in post-transcriptional circadian gene regulation. First, the paper illustrates that paraspeckles are found in pituitary GH4C1 cells and that they contain known protein components as well the long non-coding RNA Neat1. All of these components are rhythmically expressed. The authors use Neat1 FISH combined with STORM to obtain a clear view of a paraspeckle body and estimate its size. Previous studies indicated that transcripts with a IRAlu element in their 3'-UTRs are retained in paraspeckles (Chen et al. 2008). The same results were obtained here. Adding an IRAlu element to the 3'-UTR of a GFP reporter led to its retention in paraspeckles. Finally, and most importantly, retention of the GFP reporter in paraspeckles led to the oscillation of EGFP protein expression.

The experiments in this manuscript are well executed and presented. Much of the paper confirms previous results and establishes the role of paraspeckles in rat pituitary cells. The striking results of the paper are the last two figures showing that the levels of the GFP signal from the IRAlu-containing GFP reporter oscillate. This is an important experiment because it directly tests whether the retention of mRNAs in paraspeckles can drive oscillating protein expression. Additional experiments need to be done before this manuscript is acceptable for publication (see comments below).

Comments:

A) Figure 4 and Figure 5 present the key data for this paper, but they also raise several questions or concerns.

1) What drives the circadian accumulation of IRAlu-egfp in the nucleus? The authors note that paraspeckle numbers increase during the day. It also seems possible that paraspeckles size could oscillate? If technically possible, the authors should distinguish between these possibilities.

2) Figure 5 would be more convincing if the peak timepoints were assayed twice instead of once. Or average the data and provide error bars?

3) In Figure 5, only 20% of the IRAlu-GFP cells show rhythmic fluorescence. This assay was performed on those cells that showed fluorescence 10% higher than background. What was the overall% of cells showing cycling eGFP? Why is it so low?

4) A loss of function experiment is necessary to show that the IRAlu-GFP reporter does not cycle when paraspeckles are disrupted.

B) If paraspeckles can play a role in the post-transcriptional circadian gene regulation, it is important to understand the biological significance of this phenomenon. Does retention in paraspeckles drive the oscillations of known cycling transcripts? Menet et al. 2012 includes a list of putative post-transcriptional cycling transcripts. A genome wide approach should be used to determine whether known cycling transcripts are associated with paraspeckles by deep sequencing the RNA brought down in RIP experiments (Figure 2) or Neat1 RNA pull down using anti-sense oligonucleotides (Figure 4). If cycling mRNAs are found associated with paraspeckles, then a loss of function experiment could determine whether this retention contributes to cycling. These types of experiments would clearly show that paraspeckle retention plays a relevant role in circadian gene expression.

Reviewer #3:

Nuclear paraspeckles are ribonucleoprotein complexes constituted principally of the 4 proteins PSPC1, RBM14, NONO, and SFPQ and the lncRNA Neat1, and which regulate gene expression post-transcriptionally by sequestering mRNAs in the nucleus and consequently preventing translation. Leveraging on the several recent papers showing that post-transcriptional events significantly contribute to rhythmic gene expression, this paper submitted by the research team of Anne-Marie François-Bellan examines the role of paraspeckles in the regulation of circadian gene expression using mostly GH4C1 cells in vitro (a rat pituitary-derived cell line). The authors first show that the expression of the 4 major paraspeckles proteins and Neat1, as well as their interactions, are rhythmic in entrained GH4C1 cells. Based on published results showing that mRNAs containing inverted repeated Alu elements (IRAlus) are retained in paraspeckles, the authors then test if mRNAs containing these IRAlus can be subjected to paraspeckles-mediated rhythmic post-transcriptional regulation. To this end, the authors developed a strategy using egfp mRNA fused to IRAlus (or just Alus as controls), and showed that rhythmic paraspeckles expression rhythmically impacts on the nuclear retention and protein expression of egfp when fused to IRAlus.

Overall, this is a decent paper that investigates a topic that gathered special interest in the recent years in the circadian field, i.e., post-transcriptional regulation of circadian gene expression. The approaches using egfp fused with IRAlus are elegant and bring some relevant and nice mechanistic insights on how paraspeckles can contribute/regulate rhythmic gene expression. There are however quite a lot of problems with this manuscript that cannot be overcome easily, and I thus believe this paper does not meet the requirements for publication in *eLife*.

1) The post-transcriptional regulation of egfp RNAs fused to IRAlus in vitro in GH4C1 cells provides a nice mechanistic demonstration that paraspeckles impact on rhythmic gene expression. Yet, does this regulation occur in vivo on a much larger scale, or is this an oddity of the system they used? The authors could have easily used some publicly available datasets and for example assessed in the genes "labeled" as post-transcriptional cyclers harbors more IRAlus in their 3'UTR. They could even have more simply test a few of these genes in their in vitro system by comparing RNA levels between nucleoplasm and cytoplasm. This is something that, to me, is really lacking and I really think it should have been investigated to give some more breadth to the paper.

2) I have also quite a few technical comments, which should I think be addressed:a) I am a bit confused with the ICC and FISH experiments. The authors observed many dots for proteins, and only 2 dots for Neat1. Do they have an explanation for this? How was the Neat1 FISH conducted? Are they really observing paraspeckles or are these Neat1 genomic locations?b) Figure 2: The authors should use more controls for their RIP (e.g., qPCR against some other lncRNA as well as mRNA) to ensure that the antibodies are not just specially sticky to nucleic acids.c) There is a rhythm of Neat1 and its 4 associated proteins. Does this rhythm translate into a rhythm in the number of paraspeckles? In their size?d) The authors mentioned that they stably transfected GH4C1 cells with their egfp plasmids. Should not egfp be expressed in all cells, at least for the Alu-egfp?

[Editors' note: further revisions were requested prior to acceptance, as described below.]

Thank you for resubmitting your work entitled "Circadian RNA expression elicited by 3'-UTR IRAlu-paraspeckle associated elements" for further consideration at *eLife*. Your revised article has been favorably evaluated by James Manley (Senior editor), a Reviewing editor, and two reviewers.

The manuscript has been improved but there are some remaining issues that need to be addressed before acceptance, as outlined below:

In the discussion following their reviews, the reviewers agree that there are serious concerns about the Neat1 pulldown, which lacks controls. They feel that you must include a negative control in their genome-wide analysis. Possibly you have generated the data already, but did not include it in the revised version for whatever reason.

Thus, to summarize, this manuscript describes new important mechanisms that contribute to rhythmic mRNA expression, but has some problems. Addressing them (especially Neat1 pulldown controls) are necessary for publication. This should be feasible without doing any new experiments.

Reviewer #2:

The authors were able to address the recommended essential revisions as well as some of the minor points that were raised in the review process. They have shown convincingly that the retention of mRNAs in paraspeckles can drive post-transcriptional circadian oscillations both with a reporter as well as several endogenous genes. The paper is much stronger in this revised form and is now appropriate for publication in *eLife*.

Reviewer #3:

In this revised version, the authors addressed the reviewer's comments and provided additional data aiming at demonstrating that paraspeckles contribute to rhythmic gene expression at the post-transcriptional level in a pituitary cell line. In particular, the authors now include new results showing that disruption of the paraspeckles in a IRAlu-egfp cell line (using either siRNA or antisense oligonucleotides) decreases the ratio of nuclear vs. cytoplasmic egfp mRNA and impairs the rhythmic nuclear retention of egfp mRNA (Figure 5) in this cell line. The authors also show similar data for several endogenous genes (Canx, Fkbp4 and Calr; Figure 6; 4 other genes in supplementary data). While these data tend to overall support the author's conclusions, they also bring some concerns:

1) Why do six out of the seven genes exhibit in control conditions a rhythm with a period of about 20 hours?

2) Why are the effects on expression different between the siRNA and the antisense oligonucleotide?

3) Why are the phases of the nuclear/cytoplasmic mRNA ratio different between many of the genes (e.g., there is a 8-hr difference between Fkbp4 and Calr, Figure 6)? The model proposed by the authors would suggest that the nuclear retention is at its lowest for all genes when the number of paraspeckles is at its minimum, i.e., 15-hrs after medium change.

There are also some issues with the Neat1 RNA pull-down, which lacks proper controls. Assessing the specificity of the Neat1 RNA pull-down by using two independent oligonucleotide probes is indeed not sufficient/appropriate to characterize the transcripts specifically associated with Neat1. The experiment should have included -at least- a mock pull-down using a non-specific probe (luciferase, lacZ, etc.) to account for the non-specific binding of endogenous transcripts to the beads/resin. I noticed that this control is included in the methods section ("one biotinylated irrelevant probe was used for Neat1 RNA non-specific pull-down"), yet it is absent from the illumina sequencing analysis. This lack of a proper control pull-down thus significantly decreases the value of the experiment.

In conclusion, while this manuscript may appear improved, it also raises new concerns and I am therefore not sure this manuscript reaches the standard of publication in *eLife*.

---

## [Author Response]

*Essential revisions:*

*A loss of function experiment to show that the IRAlu-GFP reporter does not cycle when paraspeckles are disrupted.*

As suggested we performed new experiments to show that rhythmic nuclear versus cytoplasmic egfp mRNA in IRAlu-egfp cell line depends upon the presence of paraspeckles. Given the unique Neat1 RNA nuclear localization, RNA knockdown is more convenient than paraspeckle protein depletion for investigating paraspeckle function. Furthermore since there is an essentially perfect relationship between loss of paraspeckles and depletion of Neat1 RNA (Clemson et al., 2009), we addressed this issue by using Neat1 siRNA and Neat1 antisens oligonucleotides (ASO). By RT-qPCR we showed that Neat1 RNA levels were reduced in IRAlu-egfp cell line to around 60% after treatment with specific siRNA compared to negative control siRNA (Figure 5—figure supplement 1). The reduction we obtained in Neat1 RNA levels using siRNA is modest but comparable to that reported in human Hela cells (Clemson et al., 2009; Gagnon et al., 2014) and since in FISH experiments we do not find evidence for Neat1 RNA in the cytoplasm, this is consistent with other findings that RNA inhibition using siRNA can effectively occur in the nucleus (Langlois et al., 2005; Robb et al., 2005; Valgardsdottir et al., 2005; Clemson et al., 2009; Gagnon et al., 2014). Furthermore, the decrease in Neat1 RNA levels was not amplified when we used Neat1 ASO to disrupt paraspeckles siRNA (Figure 5—figure supplement 1). In any case, treatment with Neat1 siRNA and Neat1 ASO disrupts the circadian expression pattern of Neat1 RNA siRNA (Figure 5—figure supplement 1). More importantly, when IRAlu-egfp cells were transfected either by Neat1 siRNA or Neat1 ASO as compared to negative control, the relative ratio of nuclear versus cytoplasmic egfp mRNA levels was significantly decreased (Figure 5) and the circadian egfp nuclear retention was abolished (Figure 5). These loss of function experiments showed that the egfp mRNA nuclear retention in IRAlu-egfp cell line does not cycle when paraspeckles are disrupted.

The results are reported in the paragraph 3.5 added in the Results section and discussed in the Discussion section.

*A genome wide approach should be used to determine whether known cycling transcripts are associated with paraspeckles.*

As suggested, we performed a genome wide approach by deep sequencing the RNA brought down in Neat1 RNA pull-down using anti-sense oligonucleotides. We used the Tophat/Cufflinks pipeline (Trapnell et al., 2012) and only transcripts which exhibited values of fragment per kilobase per million of mapped reads (FPKM) higher than 1 were taken into account. The specificity of Neat 1 RNA pull-down was assessed by crossing the FPKM>1-limited lists obtained with the two specific oligonucleotides. This allowed us to provide a list of genes that were specifically associated to Neat1 RNA (see paragraph 4.1 in Results section and [Supplementary-material SD5-data]). All the RNA sequencing data have been registered at Gene Expression Omnibus (GEO) (accession no. GSE81972) and the following link has been created to allow review of record GSE81972 while it remains in private status: http://www.ncbi.nlm.nih.gov/geo/query/acc.cgi?token=ytwjsuuyppgpzmn&acc=GSE81972

We then used publicly available datasets and crossed them with our dataset. This allowed us to determine that near 19% of pituitary circadian transcripts (Hughes et al., 2007) and 27% of post-transcriptional circadian genes in the liver (Menet et al., 2012) are found associated with paraspeckles. Taking into account that rhythmic circadian genes are tissue specific and that it is unfavorable to compare datasets obtained in two different species, the robust overlap we found between our gene list and that of Hughes 2007 and Menet 2012 allows to propose that paraspeckles play a relevant role in circadian gene expression. These results have been described in the paragraph 4.2 of the Results section, illustrated in Figure 6 and discussed in the last paragraph of the Discussion section entitled “Contribution of paraspeckle nuclear retention to circadian gene expression”.

*Test a few additional genes in the* in vitro *system by comparing RNA levels between nucleoplasm and cytoplasm.*

We took advantage of our crossing analysis described above to select a few genes common to our dataset and the publicly available datasets cited above. We showed that the nuclear versus cytoplasmic mRNA ratio of these genes displayed a circadian expression pattern that is abolished when paraspeckles were disrupted either by Neat1 siRNA or by Neat1 ASO (see Figure 6, Figure 6—figure supplement 1 and [Supplementary-material SD6-data]). These losses of function experiments clearly show that paraspeckle retention of these few genes contributes to their circadian rhythmicity. These results have been described in the paragraph 4.3 of the Results section, illustrated in Figure 6, Figure 6—figure supplement 1 and [Supplementary-material SD6-data]) and discussed in the last paragraph of the Discussion section entitled “Contribution of paraspeckle nuclear retention to circadian gene expression.

[Editors' note: further revisions were requested prior to acceptance, as described below.]

*The manuscript has been improved but there are some remaining issues that need to be addressed before acceptance, as outlined below:*

*In the discussion following their reviews, the reviewers agree that there are serious concerns about the Neat1 pulldown, which lacks controls. They feel that you must include a negative control in their genome-wide analysis. Possibly you have generated the data already, but did not include it in the revised version for whatever reason.*

*Thus, to summarize, this manuscript describes new important mechanisms that contribute to rhythmic mRNA expression, but has some problems. Addressing them (especially Neat1 pulldown controls) are necessary for publication. This should be feasible without doing any new experiments.*

As noted by Reviewer #3, a non-specific oligonucleotide was actually included as negative control in our Neat1 RNA pull-down. This was indicated in the Materials and methods section and in the Results section (Figure 4 and Figure 4—figure supplement 2). This control allows to identify artifacts caused by direct, non-specific binding that are common to all affinity purification techniques. However, when performing RNA sequencing analysis, as already mentioned in our article (Results section, “Libraries were generated from the purified RNAs obtained with the two specific oligonucleotides but no library could be obtained with the non-specific oligonucleotide due to the too small quantity of material recovered”), the amount of RNA obtained with non-specific oligonucleotide was too low to generate a library. This attempt to generate a library after non-specific oligonucleotide was tried out three times without success, attesting that in our experimental conditions, direct non-specific binding of endogenous transcripts to the beads is very low.

One other source of artifacts that is more specific to hybridization capture approach is caused by hybridization events in which the capture-oligonucleotide directly hybridizes to an off-target RNA. As discussed by (Simon, 2015), an alternative method that controls for off-target RNA is to use two independent capture oligonucleotides that bind the target RNA. Specific signals are expected to be found with both oligonucleotides whereas signals that are found with only one oligonucleotide are interpreted as hybridization-induced artifacts. In our paper, we assessed the specificity of Neat 1 RNA pull-down by crossing the FPKM>1-limited lists obtained with two specific oligonucleotides.

Although the very low amounts of RNA obtained with non-specific oligonucleotide did not allow generation of library and RNA sequencing as explained above, they were still measurable by very sensitive qPCR technology in spite of very high Ct values. Thereby to convincingly show that transcripts from Neat1 RNA pull-down sequencing are specific Neat1 RNA targets, we verified by qPCR the specific association of a few of them with Neat1. To this end, we determined for the seven mRNA already selected in our paper, the enrichment obtained after Neat1 RNA pull-down with the two specific biotinylated complementary oligonucleotides that target Neat1 compared to the biotinylated irrelevant probe. As reported in the Results section and shown in a new figure supplement (Figure 6—figure supplement 1), the seven selected genes were significantly enriched after Neat1 RNA pull-down by the two specific probes compared to the non-specific probe.

*Reviewer #3:*

*In this revised version, the authors addressed the reviewer's comments and provided additional data aiming at demonstrating that paraspeckles contribute to rhythmic gene expression at the post-transcriptional level in a pituitary cell line. In particular, the authors now include new results showing that disruption of the paraspeckles in a IRAlu-egfp cell line (using either siRNA or antisense oligonucleotides) decreases the ratio of nuclear vs. cytoplasmic egfp mRNA and impairs the rhythmic nuclear retention of egfp mRNA (Figure 5) in this cell line. The authors also show similar data for several endogenous genes (Canx, Fkbp4 and Calr; Figure 6; 4 other genes in supplementary data). While these data tend to overall support the author's conclusions, they also bring some concerns:*

*1) Why do six out of the seven genes exhibit in control conditions a rhythm with a period of about 20 hours?*

*2) Why are the effects on expression different between the siRNA and the antisense oligonucleotide?*

*3) Why are the phases of the nuclear/cytoplasmic mRNA ratio different between many of the genes (e.g., there is a 8-hr difference between Fkbp4 and Calr, Figure 6)? The model proposed by the authors would suggest that the nuclear retention is at its lowest for all genes when the number of paraspeckles is at its minimum, i.e., 15-hrs after medium change.*

In the experimental design of our study, measures are performed every 4 hours. The reviewer is right to state that this time interval does not allow determining the precise value of the period and phase of the rhythm, but this was not the aim of the experiment. Our purpose was to test whether or not the seven selected genes exhibited a circadian nuclear retention (by definition with a period comprised between 20h and 28h), and whether the minimum value of this rhythm occurred roughly when the levels of paraspeckle protein components and Neat1 RNA were the lowest. We indeed found for the seven selected genes 1/ that the relative mRNA ratio between nucleoplasm and cytoplasm over the time can be fitted by a sine wave curve with a R squared > 0.55 attesting its circadian nature and 2/ that the rhythms were approximately in phase with paraspeckle component rhythms. Furthermore, while having different effects on expression levels, we also found that both siRNA and antisense oligonucleotides were able to disrupt these rhythms attesting that paraspeckles are involved in the circadian nuclear retention of these mRNA.